# ENIGMA: EEG-to-Image in 15 Minutes Using Less Than 1% of the Parameters

## Abstract

To be practical for real-life applications, models for brain-computer interfaces must be easily and quickly deployable on new subjects, effective on affordable scanning hardware, and small enough to run locally on accessible computing resources. To directly address these current limitations, we introduce **ENIGMA**, a multi-subject electroencephalography (EEG)-to-Image decoding model that reconstructs seen images from EEG recordings and achieves state-of-the-art (SOTA) performance on the research-grade THINGS-EEG2 and consumer-grade Alljoined-1.6M benchmarks, while fine-tuning effectively on new subjects with as little as 15 minutes of data. **ENIGMA** boasts a simpler architecture and requires less than 1% of the trainable parameters necessary for previous approaches. Our approach integrates a subject-unified spatio-temporal backbone along with a set of multi-subject latent alignment layers and an MLP projector to map raw EEG signals to a rich visual latent space. We evaluate our approach using a broad suite of image reconstruction metrics that have been standardized in the adjacent field of fMRI-to-Image research, and we describe the first EEG-to-Image study to conduct extensive behavioral evaluations of our reconstructions using human raters. Our simple and robust architecture provides a significant performance boost across both research-grade and consumer-grade EEG hardware, and a substantial improvement in fine-tuning efficiency and inference cost. Finally, we provide extensive ablations to determine the architectural choices most responsible for our performance gains in single-subject, multi-subject, and held-out subject transfer cases across multiple benchmark datasets. Collectively, our work provides a substantial step towards the development of practical brain-computer interface applications.

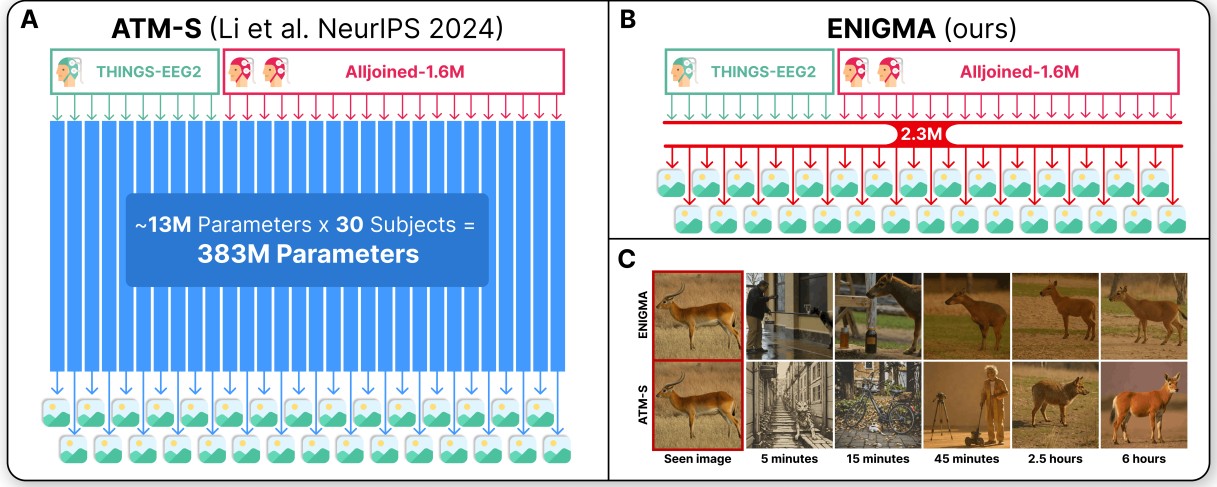

Figure 1: **A**: ATM-S (Li et al., 2024) vs **B**: **ENIGMA** (ours) comparison of model size and multi-subject capabilities on THINGS-EEG2 (Gifford et al., 2022) (green cap, 10 subjects) and Alljoined-1.6M (Xu et al., 2025) (red cap, 20 subjects) datasets. **C**: Comparison of **ENIGMA** and ATM-S across training data scale.

# 1 Introduction

Reconstructing visual experiences from brain activity has long been a goal of both neuroscience and machine learning, and a foundational step for building decoding algorithms for practical brain-computer interface (BCI) applications targeting states like mental images (Kneeland et al., 2025). Decoding visual information encoded in human brain activity could help researchers better understand cognitive processes, and could also be useful in a clinical setting (Pearson et al., 2015), where millions of patients are left unable to communicate through conventional means as a result of traumatic brain injuries, and many common afflictions manifest as a profound dysregulation of unwanted or confusing visual experiences (Holmes & Mathews, 2010). While the Natural Scenes Dataset (NSD) (Allen et al., 2022; Kneeland et al., 2025) has yielded striking functional magnetic resonance imaging (fMRI)-based reconstructions of seen images using latent diffusion models (Scotti et al., 2024), electroencephalography (EEG)-based reconstruction remains challenging due to EEG's low signal-to-noise ratio and spatial resolution. Despite these limitations, EEG remains appealing for real-time BCI applications because of its temporal precision and inexpensive, portable form factor.

Existing EEG-to-Image decoding research spans a wide range of architectural approaches: Fei et al. (2024) demonstrates a simple linear mapping from EEG to an expressive CLIP (Contrastive Language-Image Pretraining (Radford et al., 2021)) image embedding space combined with a pre-trained diffusion model, while Li et al. (2024) proposes a more complex architecture (ATM-S) utilizing a transformer-based brain encoder and a two-stage generation process utilizing the diffusion prior introduced in Scotti et al. (2023). These approaches have laid important groundwork for this field of research, however, current EEG-to-Image models face three critical barriers that prevent their deployment in practical BCI applications.

**Rapid adaptation to new subjects remains an unsolved challenge.** All existing methods for EEG-to-Image decoding (Song et al., 2024; Fei et al., 2024; Li et al., 2024) require training specialized models from scratch for each subject using hours of training data, which is impractical for real-world BCI applications that demand rapid functionality on new subjects. While recent work in fMRI-to-Image has made progress on this front—Scotti et al. (2024) reduced calibration requirements from 40 hours to 1 hour through efficient fine-tuning—even this breakthrough remains insufficient for practical deployment, as it still requires both an expensive 7 tesla fMRI scanner and a full hour of subject-specific data collection. Critically, no EEG-to-Image model has attempted to enable rapid fine-tuning on new subjects, despite EEG's advantages in accessibility and portability. We hypothesize that this gap has persisted because of EEG's noisy spatio-temporal signal characteristics, making it difficult to leverage generalizable knowledge learned from other subjects while quickly adapting to the idiosyncratic neural patterns of a new individual.

**Current approaches lack robustness when applied to affordable hardware.** The recent release of the Alljoined-1.6M dataset—designed for evaluating EEG-to-Image models on consumer-grade hardware—revealed a troubling pattern: many recent models, especially complex architectures such as ATM-S, are not robust to drops in hardware quality and fail at a much higher rate when deployed in noisier recording environments (Xu et al., 2025). This brittleness represents a fundamental obstacle to democratizing BCI technology, as the vast majority of potential users and downstream applications cannot access research-grade EEG equipment. The challenge lies in learning representations that capture the underlying neural encoding of visual information in a way that generalizes across varying levels of signal quality, electrode density, and noise characteristics. Without addressing this weakness, EEG-to-Image models will remain confined to laboratory settings, unable to translate to the accessible, consumer-grade hardware that many practical BCI applications require.

**Existing architectures are too computationally prohibitive for large-scale deployment.** Most EEG-to-Image models are quite large relative to the amount of data they are trained on, and require separate models to be trained for each subject, resulting in a linear increase in what is already a very large functional model size as models are deployed across multiple users (Song et al., 2024; Fei et al., 2024). While Li et al. (2024) demonstrated that a unified model can be trained across multiple subjects, doing so without any mechanism for handling subject-specific differences led to a substantial performance drop; attaining reasonable performance still required training separate models for each subject. These architectural and size limitations constitute a stubborn barrier for widespread deployment: running these large models on edge devices or in settings without access to server-grade GPUs is almost impossible, and supporting inference on multiple subjects requires scaling computational resources proportionally. The core difficulty lies in designing a model

architecture with performance *and* efficiency in mind, and in ensuring that a given approach can maintain subject-specific decoding performance while sharing the vast majority of parameters across subjects.

In this paper, we introduce **ENIGMA** (**E**EG **N**eural **I**mage **G**enerator for **M**ulti-subject **A**pplications), a multi-subject model for reconstructing seen images from EEG data that directly addresses each of these three critical weaknesses. Our work includes several notable contributions:

**(1) ENIGMA** is the first EEG-to-Image model to quickly fine-tune on new subjects with as little as 15 minutes of data, making it effective for practical downstream use cases (Figure 1C). **(2)** Our model achieves robust performance across hardware quality levels, producing SOTA performance on both available benchmark datasets, and a substantial performance gain on Alljoined-1.6M, collected on consumer-grade hardware. **(3) ENIGMA** is unified across subjects and datasets, requiring less than 1% of the trainable parameters to decode the 30 subjects across THINGS-EEG2 and Alljoined-1.6M datasets (5.5x reduction vs ATM-S $\times$ 30 subjects = 165x reduction in total parameters) when compared to single subject approaches (Figure 1AB). **(4)** We are the first EEG-to-Image work to provide extensive evaluations using behavioral experiments with human raters, a standard in the adjacent field of fMRI-to-Image research, and we conduct a detailed ablation analysis investigating which architectural aspects are most effective across differences in hardware quality and multi-subject configurations.

Our work demonstrates significant progress towards overcoming the three fundamental barriers that have prevented EEG-to-Image decoding from practical deployment. By enabling rapid fine-tuning with minimal data, maintaining robust performance across hardware configurations, and achieving dramatic improvements in parameter efficiency, **ENIGMA** represents a sizable step on the path to bringing real-time, accessible EEG-to-image decoding from laboratory demonstration to practical clinical, consumer, and research applications.

## 2    Related Work

**fMRI-to-Image Reconstruction.** The advent of generative diffusion models has revolutionized neural decoding for adjacent scanning modalities like fMRI (Ozcelik & VanRullen, 2023; Scotti et al., 2023; 2024; Takagi & Nishimoto, 2023a;b; Kneeland et al., 2023b;a;c). Takagi & Nishimoto (2023a) demonstrated some of the first high-resolution image reconstructions from fMRI by mapping fMRI activity into the latent space of a diffusion model. Ozcelik & VanRullen (2023) were the first to show that latent diffusion can reconstruct natural scenes from fMRI with high semantic fidelity, and defined a set of evaluation metrics combining low-level (pixel-wise) and high-level (feature-based) similarity measures that have become standard (Ozcelik & VanRullen, 2023; Scotti et al., 2023; 2024; Takagi & Nishimoto, 2023a;b; Kneeland et al., 2023b;a;c). While fMRI enables finer-grained reconstructions due to its high spatial resolution, EEG's superior temporal resolution and portability makes it more suited for real-time applications, despite its lower signal fidelity.

**Multi-Subject fMRI Models.** Recent work in fMRI-to-Image decoding has demonstrated the effectiveness of multi-subject pretraining for rapid adaptation to new subjects. Scotti et al. (2024) introduced MindEye2, which employs a shared-subject functional alignment approach where subject-specific ridge regression linearly maps each individual's fMRI voxel patterns to a common 4096-dimensional latent space, followed by a shared non-linear pipeline across all subjects. By pretraining on 7 subjects and fine-tuning on new subjects, MindEye2 achieves high-quality reconstructions with as little as 1 hour of calibration data, a 40x reduction compared to single-subject approaches. However, such linear alignment strategies on the input data do not translate well to EEG-to-Image decoding due to fundamental differences in signal characteristics: EEG exhibits complex spatio-temporal dynamics with highly variable signal topology across subjects, lower spatial resolution, and substantially higher noise levels compared to the relatively stable spatial patterns of fMRI voxel activity. These challenges necessitate different architectural approaches for achieving robust multi-subject EEG decoding, which we address in this work.

**EEG-based Visual Decoding.** Decoding visual content from EEG recordings uses a wide array of approaches spanning classification, retrieval, and image reconstruction. While early studies collected EEG recordings in response to visual stimuli (Spampinato et al., 2017), many contained confounds that made it difficult to decode true semantic content (Li et al., 2020), highlighting a need for better data and methods. This critique, along with improvements in EEG preprocessing and experimental design, led to the development of new datasets as part of the THINGS initiative. Grootswagers et al. (2022) released THINGS-EEG (50

subjects, 1,854 concepts) using rapid serial visual presentation, and Gifford et al. (2022) further improved upon THINGS-EEG with THINGS-EEG2, emphasizing trial randomization and quality control, which has since become a standard benchmark for EEG vision decoding. On THINGS-EEG2, new methods (Song et al., 2024; Li et al., 2024) employing contrastive learning between image and EEG features have achieved significant gains in zero-shot object classification. Alljoined-1.6M (Xu et al., 2025) extends this paradigm to twice as many subjects and to a consumer-grade EEG hardware setup, providing a new set of tools for developing and evaluating EEG-based visual decoding methods for practical use.

**EEG-to-Image Reconstruction.** In the space of EEG-to-Image reconstruction methods, Li et al. (2024) introduced a specialized EEG encoder called the Adaptive Thinking Mapper (ATM-S), which uses a two-stage decoding approach comprising a transformer, a CNN, an MLP, and a diffusion prior before using the decoded CLIP vector to generate an image reconstruction using a diffusion model. The complexity of the ATM-S architecture requires careful tuning of many intricate architectural components and multiple sequential training stages to be successful. It was also shown with the release of the Alljoined-1.6M dataset that this degree of complexity results in brittle performance on lower grade EEG hardware (Xu et al., 2025). While the architecture does support multi-subject training through a learned subject embedding in the transformer stage, training ATM-S on multiple subjects produces a substantial performance drop, so for most analyses in this work we use the model in its single-subject configuration.

Inspired by Ozcelik & VanRullen (2023), Perceptogram (Fei et al., 2024) utilizes a linear transform to map EEG recordings to a CLIP embedding space, and generates images directly from the predicted embeddings using a diffusion model. While its reconstructions still contain less detail than fMRI-based reconstruction, the approach of (Fei et al., 2024) demonstrates the significant power of robust linear models in producing recognizable images from low SNR brain activity patterns.

# 3 ENIGMA

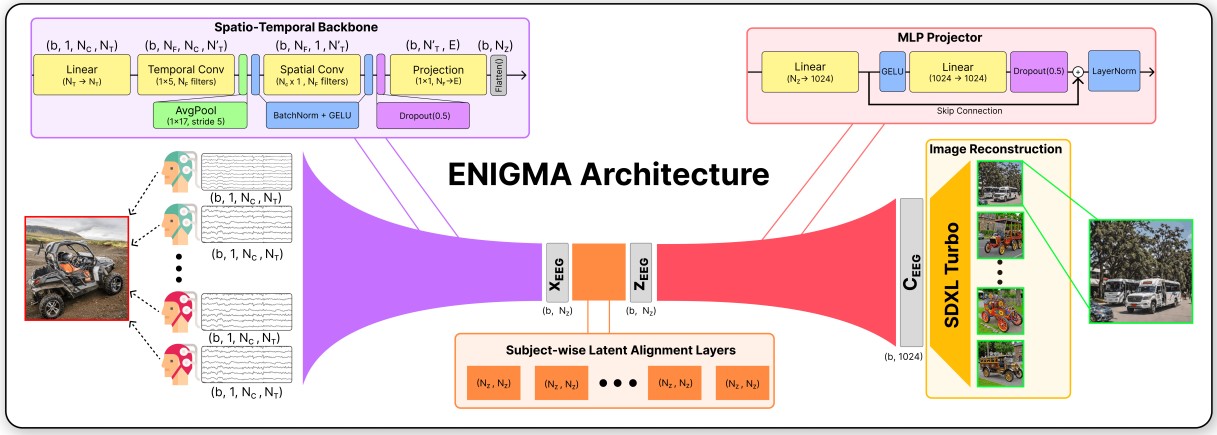

Figure 2: During training, brain activity from each subject is passed through a shared pathway of spatio-temporal convolutions, producing an intermediate latent vector $x_{\text{EEG}}$ from all subjects. This is then passed through a set of subject-specific latent alignment layers to produce an aligned latent embedding $z_{\text{EEG}}$. This latent is passed through a fully connected MLP projection layer to produce the output $c_{\text{EEG}}$ vector, which is reconstructed into an image using SDXL. Details of these procedures are provided in Section 3.3.

## 3.1 Methodology

We designed our model to adhere to several key requirements for practical BCI applications: **(1)** high performance on both research-grade and consumer-accessible EEG hardware, **(2)** a subject-unified model architecture that rapidly fine-tunes on new subjects, and **(3)** a small, scalable, and efficient design that minimizes model complexity and compute requirements for both training and inference.

To meet these requirements, our proposed model, **ENIGMA**, has 4 components: **(1)** a spatio-temporal convolutional neural network to learn a set of robust features from the spatial and temporal dimensions of the input signal, **(2)** a set of subject-wise latent alignment layers to capture and navigate subject-specific differences in the latent space of the model, **(3)** an MLP projector to the ViT-H/14 CLIP (Radford et al., 2021) embedding space, and **(4)** a pretrained diffusion model (i.e. SDXL Turbo) that can invert a CLIP embedding into an image.

Unlike previous approaches, our model can operate in three distinct modes: single-subject (trained on one subject at a time), multi-subject (trained on multiple subjects in parallel using shared model weights), and fine-tuned (pretrained on one or more subjects, and fine-tuned on a target subject). In practice **ENIGMA** has near identical performance in the single-subject and fine-tuned conditions when using all available training data (see Figure 4B), and so we primarily report performance metrics in the single and multi-subject configurations, while examining the fine-tuning efficiency benefits provided by our pretraining approach by reporting performance using only 15 minutes of training data.

### 3.2  Datasets

**THINGS-EEG2** (Gifford et al., 2022) is the current public benchmark for EEG-based visual decoding. It contains 64-channel ActiChamp (costing $\sim$\$60,000) recordings at 1000 Hz from 10 participants viewing 16740 unique images in a rapid serial visual-presentation paradigm. 200 of these images are designated as the testing set and are repeated 80 times each, while the remaining 16540 images in the training set are repeated 4 times each, for a total of $\sim$820k trials across the whole dataset.

**Alljoined-1.6M** (Xu et al., 2025) is a follow-up corpus of EEG responses to visual stimuli collected with a much cheaper consumer-grade 32-channel Emotiv Flex 2 gel headset ($\sim$\$2,200) at 250 Hz comprising the same stimuli and experimental paradigm as THINGS-EEG2, across 20 new subjects, for a total of $\sim$1.6M trials. Here we use "consumer-grade" to denote an affordability and accessibility tier (low cost, portable) relative to research-grade systems such as the 64-channel ActiChamp, rather than to imply that such hardware is unsuitable for research. Emotiv devices have been validated for EEG and ERP research (Badcock et al., 2013; Williams et al., 2020).

Both datasets are processed at 250 Hz; THINGS-EEG2 is recorded at 1 kHz and resampled to 250 Hz to match Alljoined-1.6M (Appendix A.1). Other preprocessing steps are in Appendix A.1. When reproducing other methods, we follow their preparation and preprocessing steps.

### 3.3  Architecture

The following components of our **ENIGMA** architecture are depicted in Figure 2.

**Spatio-Temporal Backbone** The preprocessed EEG data is passed through an embedding module that applies convolutions over the temporal and spatial dimensions of our data. We treat multichannel EEG with $N_C$ channels and $N_T$ time points as an "image" of shape $1 \times N_C \times N_T$.

A temporal 2D convolution (kernel $(1,5)$, $N_F = 40$ feature maps) is followed by average pooling over time (kernel $(1,17)$, stride 5), batch normalization (BN), and a GELU non-linearity:

$$z_1 = \text{Conv2D}_{1 \times 5}^{N_F}(EEG)$$
$$z_2 = \text{AvgPool}_{1 \times 17, 5}(z_1)$$
$$h_1 = \text{GELU}(\text{BN}(z_2))$$

where $h_1 \in \mathbb{R}^{N_F \times N_C \times N_T'}$ with $N_T' = \left\lfloor \frac{N_T - 21}{5} \right\rfloor + 1$ (i.e., $N_T = 250 \Rightarrow N_T' = 46$). Next, a spatial convolution (kernel $(N_C, 1)$, $N_F$) integrates information across channels, followed by BN and GELU:

$$z_3 = \text{Conv2D}_{N_C \times 1}^{N_F}(h_1)$$
$$h_2 = \text{GELU}(\text{BN}(z_3))$$

where $h_2 \in \mathbb{R}^{N_F \times 1 \times N_T'}$. At 250 Hz each time point spans 4 ms, so the temporal convolution (kernel size 5) applies a 20 ms window. The subsequent average pooling (kernel 17, stride 5) yields $N_T' = 46$ temporal bins:

each bin has an effective receptive field of roughly 84 ms, and successive bins are spaced 20 ms apart, so the sequence of bins preserves temporal structure across the full post-stimulus window. This range is well matched to the timescales relevant for visual EEG decoding: the 20 ms kernel resolves fast transients, the roughly 84 ms pooled window is comparable to the width of early visual evoked components (on the order of 100-200 ms), and the sequence of bins captures slower oscillatory dynamics in the theta and alpha bands. We leave a systematic ablation of the temporal kernel size and pooling stride to future work.

We then apply dropout (Srivastava et al., 2014) ($p = 0.5$) for regularization and project the features to embedding dimension $E_p = 4$ with a $1 \times 1$ convolution. Finally, we flatten the output $\in \mathbb{R}^{E \times 1 \times N'_T}$ to obtain a latent embedding vector $X_{EEG} \in \mathbb{R}^{N_z}$ of dimension $N_z = 184$.

**Subject-wise Latent Alignment Layers** To account for systematic differences between subjects, we learn a set of subject-specific fully-connected linear alignment layers with weights $W_s \in \mathbb{R}^{N_z \times N_z}$ that output an aligned latent representation $z_{EEG}$ across subjects. This alignment module allows downstream modules to learn a unified mapping to the CLIP embedding space across subjects, and is an important piece in allowing the vast majority of model parameters to be shared across multiple subjects with diverse signal dynamics.

**MLP Projector** After obtaining the aligned $z_{\text{EEG}}$ latent embedding, we use a projection head to map it to the final CLIP ViT-H/14 latent dimension $D = 1024$. The head is a feed-forward MLP network with a skip connection: a linear layer from 184 to 1024, followed by GELU and dropout, then another linear layer, and finally layer normalization. The residual is added to the output of the second linear layer before normalization, to help stabilize training and allow the model to refine the initial linear projection with non-linear adjustments. The module outputs an EEG-predicted CLIP embedding $c_{\text{EEG}} \in \mathbb{R}^{1024}$.

**Image Reconstruction** To generate images from the EEG embedding $c_{\text{EEG}}$, we leverage Stable Diffusion XL Turbo (SDXL) (Sauer et al., 2024), and its associated CLIP ViT-H/14 image-prompt adapter (IP-Adapter) (Ye et al., 2023). The IP-Adapter is a lightweight module inserted into SDXL's cross-attention layers, which enables an image embedding to steer image generation alongside an optional text prompt. We optimize our model to predict the CLIP ViT-H/14 image embeddings expected by SDXL Turbo's IP-adapter as input. Formally, SDXL solves:

$$x_T \sim \mathcal{N}(0, I), \tag{1}$$

$$x_{t-1} = f_\theta(x_t, c_{\text{text}}, c_{\text{EEG}}, t) + \text{noise} \tag{2}$$

for $t = T, T-1, \ldots, 0$, where $c_{\text{text}}$ is the text context (which in our case is an unconditional placeholder embedding) and $c_{\text{EEG}}$ is our injected EEG image embedding. We run the diffusion for 4 inference steps, which is standard for this version of SDXL.

**Loss Functions and Training** Following Li et al. (2024), we align the EEG embedding $c_{\text{EEG}}$ to the CLIP ViT-H/14 image embedding of the stimulus image $f_{\text{CLIP}}(\text{image}) \in \mathbb{R}^{1024}$ by minimizing the Mean-Squared Error (MSE) between the two and regularizing with the InfoNCE contrastive loss (van den Oord et al., 2019; Radford et al., 2021). The former matches the EEG embedding to its corresponding image embedding in CLIP latent space, and the latter ensures that the embedding retains relevant directional semantics within the CLIP manifold, while learning to discard the subject and session-specific information. The InfoNCE term uses symmetric in-batch negatives, averaging the EEG-to-image and image-to-EEG contrastive directions (Appendix A.2). The relative weight of these two losses is modulated by $\lambda = 0.5$. Our overall loss function can be seen in Equation 3, and the full mathematical formulations of our loss functions can be found in Appendix A.2.

**Inference** All fitted preprocessing parameters, namely the per-channel batch-normalization statistics and the whitening matrix $W = \Sigma^{(-1/2)}$ (with $\Sigma$ the noise covariance estimated on the training partition), are fit once offline and then frozen, so at inference each is applied as a fixed per-trial operator: batch normalization runs in evaluation mode using its stored running statistics, and whitening is the linear map $x$ to $Wx$. The global z-scoring (per-channel mean and standard deviation) is likewise fit on the training split only. The pipeline therefore requires no test-set statistics, no future trials, and no batch, and runs on a single incoming epoch. The one non-causal operation is the zero-phase (filtfilt) bandpass filter, which draws on other samples within the analysis epoch; the system is thus online at the granularity of a buffered epoch (a fixed latency on the order of the epoch length) rather than strictly sample-by-sample causal streaming. Repetition averaging (Section

4.1) is an inference-time aggregation and is omitted in a single-trial online setting, at the signal-to-noise cost characterized in Appendix A.9.

$$\mathcal{L} = \text{MSE}(c_{EEG}, f_{CLIP}(\text{image})) \quad + \lambda \cdot \text{InfoNCE}(c_{EEG}, \text{norm}(f_{CLIP}(\text{image}))) \tag{3}$$

**ENIGMA** was trained in FP32 on an RTX 3090 GPU using the AdamW optimizer with learning rate 3e-4 and batch size 512. In the multi-subject configuration, we train for 150 epochs on the training split of both THINGS-EEG2 and Alljoined-1.6M simultaneously (30 subjects, ∼2M data points averaged for each image to ∼500k training trials). In this configuration, the model takes 5.5 hours to train across all 30 subjects, and 10 minutes to train on a single subject. We note that in practice our model could also be trained on GPUs with as little as 8GB of VRAM and fine-tuned in as little as 2 minutes when using a 15 minute calibration period. Such minimal computational requirements allow **ENIGMA** to be rapidly deployable on edge computing devices, facilitating a wide array of downstream applications.

## 4 Results

We report results on two of the most recent and prominent EEG-to-Image benchmarks: THINGS-EEG2 (Gifford et al., 2022) and Alljoined-1.6M (Xu et al., 2025), and evaluate **ENIGMA** against the only available EEG-to-Image baselines for those datasets, Perceptogram (Fei et al., 2024) and ATM-S (Li et al., 2024). Figure 3 presents a set of the best reconstructed images, comparing our method with available baselines on both available datasets. These examples illustrate typical outcomes: our reconstructions generally capture the correct high-level object, e.g., oranges, sheep, furniture, etc. Perceptograms are usually blurrier and sometimes miss the object entirely, e.g., producing a vague shape or significant visual distortions. The ATM-S images are categorically similar, but are often less visually specific to the precise object being decoded. Examples of median and worst-case reconstructions can also be seen in Appendix A.5.

### 4.1 Quantitative Evaluations

Table 1 summarizes the quantitative performance of **ENIGMA** and baseline methods on both benchmarks in single-subject, multi-subject, and held-out subject transfer configurations. For all methods, we output 10 reconstructions per test sample from each method and report averaged image feature metrics across them. For multi-subject configurations, **ENIGMA** achieves the best scores on all metrics, indicating that the latent alignment layers are enabling our model to align representations across subjects from both datasets. For single-subjects, our model still provides SOTA performance on the majority of metrics.

To evaluate reconstruction quality we adopt the image-similarity suite standardized in the fMRI-to-Image literature (Ozcelik & VanRullen, 2023; Scotti et al., 2023; 2024) which lets our results be read against the conventions of that adjacent field. The metrics fall into four groups. Low-level metrics capture pixel and structural fidelity: PixCorr is the pixel-wise correlation between the stimulus and reconstruction, and SSIM is the structural similarity index (Wang et al., 2004). High-level metrics capture semantic and feature-space agreement using pretrained vision models: AlexNet(2) and AlexNet(5) are two-way comparisons (2WC) at layers 2 and 5 of AlexNet, Incep is the 2WC at the final pooling layer of InceptionV3, and CLIP is the 2WC at the output layer of CLIP ViT-L/14, while EffNet-B (Eff) and SwAV are feature-space distances for which lower is better. A two-way comparison reports how often the stimulus embedding is closer to its target reconstruction than to a randomly drawn distractor reconstruction, so chance is 50%. Retrieval metrics (Top-1, Top-5, Top-10) measure how often the correct image is recovered from the test pool given the predicted embedding. Finally, human-rater identification accuracy (Section 4.2) measures whether human observers can match a reconstruction to its stimulus in a two-alternative forced-choice task. We compute and report the same metrics across all methods to enable direct comparison; full definitions are in Appendix A.3. Because all methods here share the same frozen image generator, the low-level metrics (PixCorr, SSIM) are strongly shaped by the generator's natural-image prior: a randomly initialized encoder attains comparable values (Table 1, ENIGMA (Random-init)), so we treat these as weak indicators of decoding fidelity and assess reconstruction quality primarily from the high-level semantic, retrieval, and human-rater metrics.

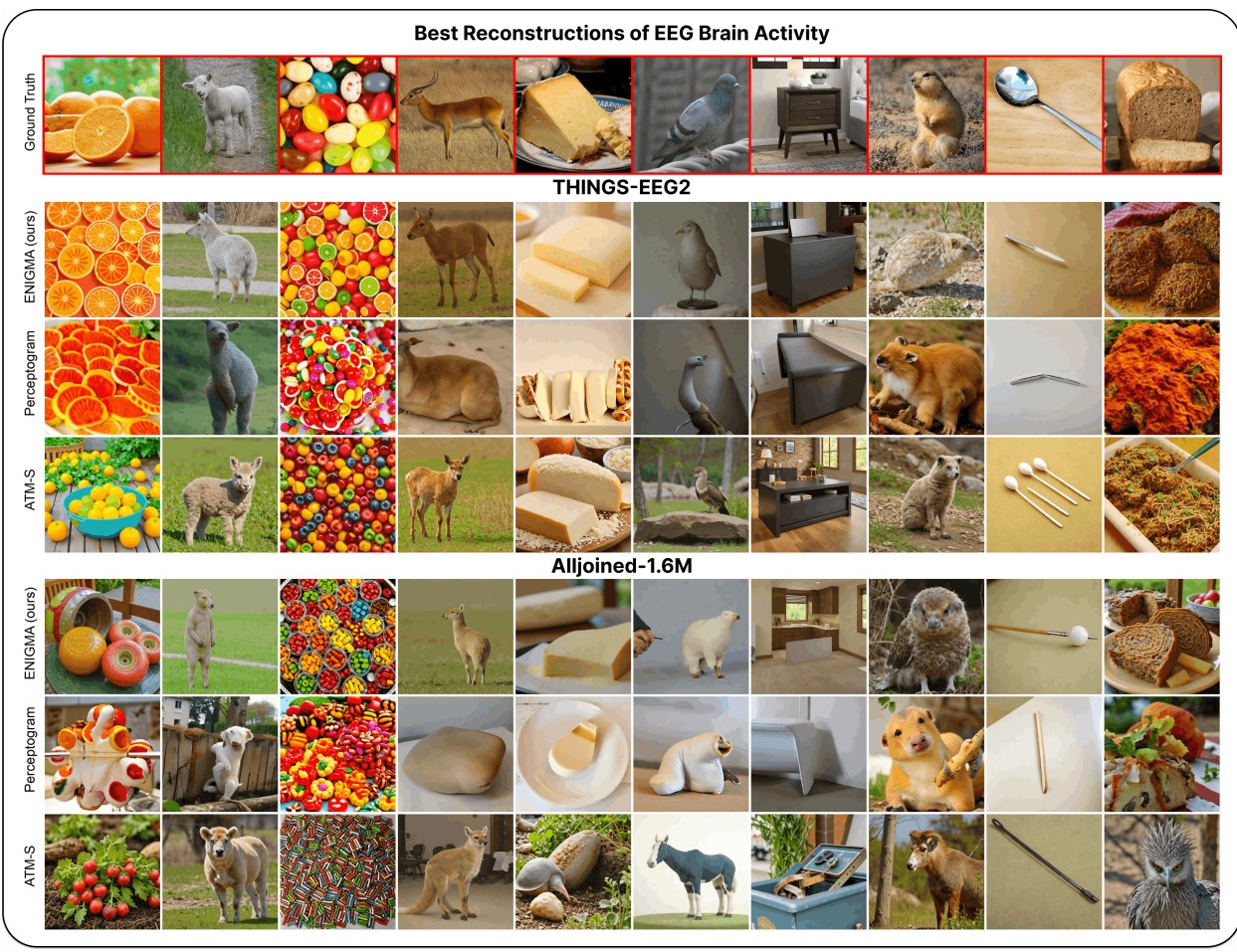

Figure 3: Qualitative comparison of reconstruction methods on seen stimuli from THINGS-EEG2 and Alljoined-1.6M. Reconstructions selected are the outputs sampled from each method and stimulus with the highest scores on all of the image feature metrics in Table 1.

| Method | Model Properties | | Low-Level | | High-Level | | | | | | | Retrieval | | | Human Raters |
|---|---|---|---|---|---|---|---|---|---|---|---|---|---|---|---|
| | # of Parameters ↓ | Inference GFLOPS ↓ | PixCorr ↑ | SSIM ↑ | Alex(2) ↑ | Alex(5) ↑ | Incep ↑ | CLIP ↑ | Eff ↓ | SwAV ↓ | Top-1 ↑ | Top-5 ↑ | Top-10 ↑ | Ident. Acc. ↑ |
| **THINGS-EEG2 (10 subjects)** | | | | | | | | | | | | | | |
| **ENIGMA (Multi-Subject)** | **2,376,842** | **294.4** | **0.1750** | **0.4313** | **84.09%** | **89.90%** | **77.98%** | **81.76%** | **0.8450** | **0.5336** | **30.20%** | **62.00%** | **72.60%** | **86.04%** |
| ATM-S (Multi-Subject) | 12,815,311 | 3,858.6 | 0.072 | 0.403 | 57.09% | 58.99% | 52.86% | 55.04% | 0.963 | 0.663 | 16.20% | 45.10% | 62.20% | 56.82% |
| **ENIGMA (Single-Subject)** | 13,896,820 | 294.4 | 0.1687 | 0.4153 | 83.19% | 89.08% | 77.33% | 81.19% | 0.8644 | 0.5470 | 31.15% | 63.00% | 76.50% | 86.82% |
| **ENIGMA (15m Fine-tune)** | 13,896,820 | 294.4 | 0.1108 | 0.4018 | 71.90% | 81.11% | 67.62% | 71.67% | 0.9251 | 0.6004 | 10.05% | 30.10% | 43.25% | – |
| ATM-S (Single-Subject) | 128,153,110 | 3,858.6 | 0.136 | 0.392 | 73.85% | 80.83% | 67.56% | 71.28% | 0.909 | 0.601 | 30.15% | 60.15% | 73.60% | 77.14% |
| Perceptogram (Single-Subject) | 4,731,924,800 | 2,807.8 | **0.247** | **0.431** | **85.46%** | 88.03% | 70.40% | 71.98% | 0.902 | 0.581 | – | – | – | 79.17% |
| **ENIGMA (Random-init)** | 13,896,820 | 294.4 | 0.1462 | 0.4240 | 49.43% | 49.46% | 49.21% | 48.90% | 0.9803 | 0.7219 | 0.50% | 2.50% | 5.00% | – |
| **Alljoined-1.6M (20 subjects)** | | | | | | | | | | | | | | |
| **ENIGMA (Multi-Subject)** | **2,376,842** | **588.8** | **0.0854** | **0.4197** | **68.24%** | **73.33%** | **63.19%** | **66.44%** | **0.9222** | **0.6082** | **8.90%** | **25.15%** | **34.27%** | **70.74%** |
| ATM-S (Multi-Subject) | 12,765,711 | 7,717.2 | 0.068 | 0.417 | 53.49% | 53.36% | 50.72% | 51.46% | 0.965 | 0.668 | 0.72% | 4.12% | 7.55% | 52.18% |
| **ENIGMA (Single-Subject)** | 27,793,640 | 588.8 | 0.0851 | 0.4153 | 68.42% | 73.48% | 63.09% | 66.33% | 0.9297 | 0.6171 | 8.50% | 24.68% | 36.77% | 71.82% |
| **ENIGMA (15m Fine-tune)** | 27,793,640 | 588.8 | 0.0429 | **0.4311** | 53.31% | 54.36% | 51.79% | 56.37% | 0.9710 | 0.6435 | 1.28% | 5.55% | 9.60% | – |
| ATM-S (Single-Subject) | 255,314,220 | 7,717.2 | 0.090 | 0.374 | 55.91% | 58.25% | 54.07% | 56.25% | 0.960 | 0.673 | 1.76% | 7.21% | 11.91% | 60.31% |
| Perceptogram (Single-Subject) | 9,463,849,600 | 5,615.6 | **0.094** | 0.401 | 67.36% | 69.28% | 58.18% | 59.94% | 0.945 | 0.637 | – | – | – | 62.00% |
| **ENIGMA (Random-init)** | 27,793,640 | 588.8 | 0.1503 | 0.4281 | 49.53% | 49.16% | 49.21% | 49.05% | 0.9807 | 0.7233 | 0.53% | 2.53% | 4.98% | – |

Table 1: Comparison of EEG-to-Image reconstruction models on the THINGS-EEG2 and Alljoined-1.6M datasets via image similarity metrics. Parameter counts and GFLOPS are computed by adding up the number of parameters and compute necessary to decode all subjects in each dataset (10 subjects for THINGS-EEG2, 20 for Alljoined-1.6M) in each configuration (single vs multi-subject). The "ENIGMA (15m Fine-tune)" rows report ENIGMA pretrained and then fine-tuned on 15 minutes of each target subject's data. Details on the human identification accuracy metric are provided in Section 4.2 and Appendix A.8. For the # of parameters, GFLOPS, EffNet-B, and SwAV, lower is better. For all other metrics, higher is better. Bold indicates best performance, and underlines second-best performance. Additional details on the metrics are in Appendix A.3.

## 4.2 Human Behavioral Evaluations

For brain decoding models to be deployed in BCI applications, their outputs must be meaningfully interpretable to users, scientists, and clinicians. While many of the metrics in Table 1 are commonly used as proxies for perceptual quality, prior research has established that these automated metrics often fail to align with human assessments of content (Sinha & Russell, 2011) or quality (Kirstain et al., 2023). Human judgment of reconstruction quality is therefore an essential performance metric for evaluating EEG-to-Image models.

To address this, we conducted a carefully controlled, large-scale online behavioral experiment where human raters (n = 545) assessed the quality of reconstructions through 2-alternative forced choice judgments. In each trial, raters determined whether a reconstruction was more similar to its corresponding ground truth image than to a randomly selected reconstruction from the same method, dataset, and subject. This task evaluates whether reconstructions contain meaningful stimulus-specific content, a minimum requirement for practical utility. Detailed experimental protocols are provided in Appendix A.8.

Our results in Table 1 confirm that **ENIGMA** achieves a substantial improvement in human identification accuracy across all evaluated conditions, which we believe to be the most reliable indicator of performance on EEG-to-Image tasks. We also confirm that all reconstruction models evaluated here produce above-chance performance in our randomized control trials ($p < 0.001$).

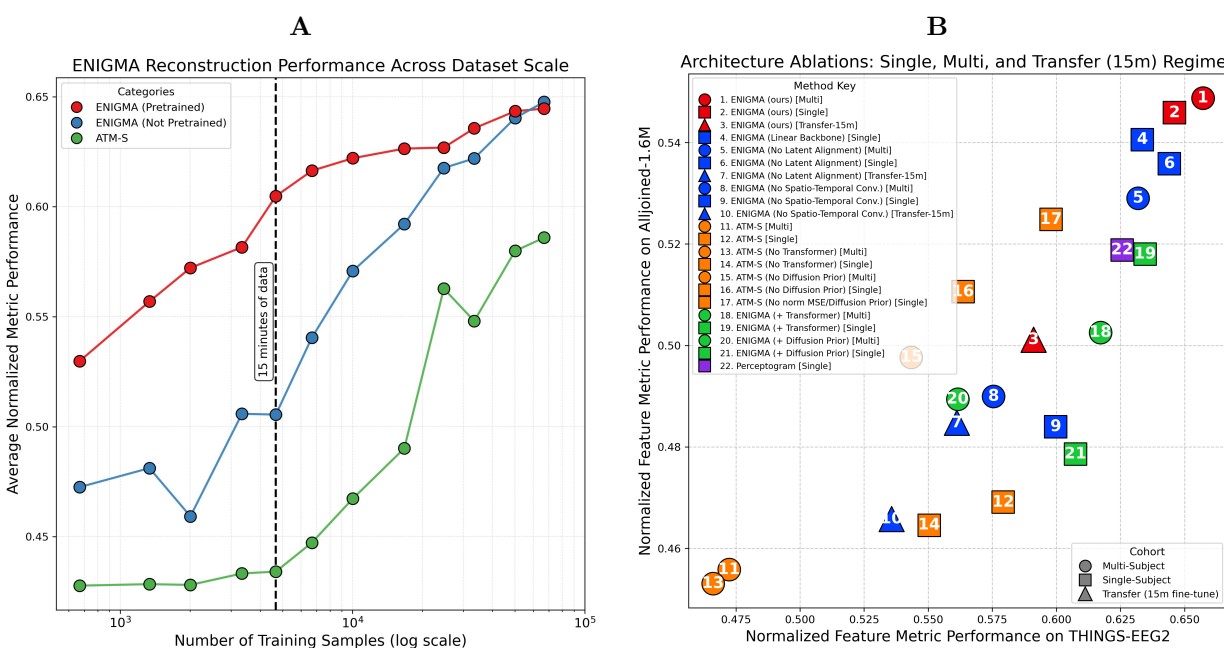

Figure 4: **(A)** Scaling efficiency of **ENIGMA** with (red) and without (blue) pretraining on other subjects, and ATM-S (Li et al., 2024) (green). Performance is plotted using varying amounts of target subject training/fine-tuning data on a log-scale X-axis. Reconstruction accuracy, evaluated using the normalized average of feature metrics presented in Table 1, is plotted on the Y-axis. All metrics are calculated on the median subject (2) of the THINGS-EEG2 dataset. **(B)** Ablation analyses: model variants (numbered icons) in single (square), multi-subject (circle), and 15m fine-tuned (triangle) configurations under each ablation type (color) are assessed via the normalized average of all feature metrics (Table 1), with THINGS-EEG2 performance on the x-axis and Alljoined-1.6M performance on the y-axis.

## 4.3 Fine Tuning Efficiency

One of the primary goals of our work is to enable high-quality reconstructions from brief calibration sessions suitable for clinical, consumer, and research contexts where extended data collection is infeasible. Figure 4A demonstrates this capability by comparing reconstruction performance across **ENIGMA** models trained or fine-tuned on varying amounts of subject-specific data. The vertical dashed line marks a 15-minute

calibration threshold, a reasonable constraint for real-world BCI deployment. While both pretrained and non-pretrained models improve consistently as training data increases, multi-subject pretraining provides substantial advantages in the low-data regime. At the 15-minute threshold (approximately 4,000 training samples), pretrained **ENIGMA** surpasses the fully-trained ATM-S architecture, while the non-pretrained baseline fails to produce identifiable reconstructions. We verify that this threshold is not an artifact of the median subject: across all 30 subjects, every subject's pretrained fine-tune surpasses fully-trained ATM-S, with a median calibration time of 19.4 minutes (IQR 4.8 to 36.4), 16.6 minutes on THINGS-EEG2 and 19.5 minutes on Alljoined-1.6M (Appendix A.11). Further analysis of the scaling performance on each benchmark dataset and of the number of EEG channels can also be found in Appendices A.6.1 and A.7. These results provide empirical evidence that **ENIGMA** and its multi-subject latent alignment procedure enable effective transfer learning in EEG-to-Image decoding, addressing a critical gap where no prior work has demonstrated functional performance on a novel subject with such limited calibration data.

### 4.4 Ablation Study

To rigorously evaluate EEG-to-Image architectures and critically examine why our method succeeds in multi-subject and consumer-grade contexts where ATM-S breaks down, we conducted ablations on key design choices in our method. Colored numeric identifiers refer to the ablation results in Figure 4B. **ENIGMA** is displayed as (1), [2], and /3\, (multi-subject, single-subject, and 15-minute transfer to a held-out subject, respectively).

**ENIGMA Modules.** While linear models do not provide any of the parameter efficiency or fine-tuning benefits of our **ENIGMA** architecture, we find using a well regularized linear backbone [4] in single-subject contexts remains an effective way to decode semantic information from brain activity (Ozcelik & VanRullen, 2023; Fei et al., 2024). We also find that eliminating the latent alignment layer (5), [6] harms performance disproportionately in multi-subject contexts, showing that the module specifically helps drive cross-participant generalization. Removing the spatio-temporal convolution stack decreases accuracy in all contexts (8), [9], confirming that joint space-time feature extraction is essential for capturing the semantic information encoded in EEG brain activity.

**Transfer-Regime Ablation.** Because **ENIGMA**'s central claim concerns transfer to new subjects, we additionally evaluate the load-bearing components in the pretrained-then-fine-tuned-to-a-held-out-subject regime (triangles in Figure 4B), using a 15-minute calibration set. We focus on the two components most relevant to transfer: the per-subject latent alignment layers (the explicit cross-subject transfer mechanism) and the shared spatio-temporal backbone (the component pretrained across subjects). The full **ENIGMA** transfer model /3\ outperforms both ablations on both datasets. Removing the latent alignment layers /7\ degrades transfer performance substantially, and far more than the same ablation affects the within-subject single-subject model (compare [6] to [2], which is nearly unchanged), confirming that these layers specifically enable cross-subject transfer rather than only within-training-subject performance. Removing the shared spatio-temporal backbone /10\ is the most damaging variant in the transfer regime, consistent with its role as the pretrained, cross-subject feature extractor.

**ATM-S Modules.** While ATM-S's (11), [12] diffusion prior slightly improves its performance on THINGS-EEG2 [16], we find in our analysis that removing it significantly improves performance on data using the consumer-grade EEG hardware in Alljoined-1.6M, and on both datasets in multi-subject contexts (15). We also noticed that if we remove the normalization of the target vectors in the MSE loss computation (see Appendix A.2 for details), results improve even further and negate the need for the diffusion prior entirely [17]. We observe similar patterns with the transformer encoder (13), [14], which only benefits narrow single-subject THINGS-EEG2 settings. These results suggest that these specific architectural complexities hinder both robustness to lower-SNR data and the ability to capture subject-specific variations in unified multi-subject models, informing our simple and robust approach.

**ENIGMA + ATM-S Modules.** To evaluate whether the architectural modules introduced by Li et al. (2024) would be beneficial to **ENIGMA**, we grafted ATM-S's transformer-based encoder and diffusion prior training stage onto **ENIGMA** (18),[19],(20),[21]. We find that both of these modules harm performance on both datasets in both single and multi-subject contexts, highlighting how our simple and robust design

captures all of the necessary information encoded in brain activity without needing additional computationally expensive modules.

# 5 Discussion

Here we introduce **ENIGMA**, a multi-subject EEG-to-Image reconstruction model that directly addresses three critical barriers preventing practical deployment of visual decoding systems: the inability to rapidly fine-tune on new subjects, brittle performance on consumer-grade hardware, and computational prohibitiveness at scale. Through multi-subject latent alignment and a suite of other techniques, **ENIGMA** achieves performant reconstructions with only 15 minutes of calibration data, an order of magnitude reduction compared to existing approaches that require hours of subject-specific training. Our lightweight spatio-temporal CNN architecture maintains robust performance across both research-grade (THINGS-EEG2) and consumer-grade (Alljoined-1.6M) hardware, demonstrating that architectural simplicity and multi-subject pretraining enable effective performance across signals of varying quality. Finally, by sharing model weights across subjects with only lightweight subject-specific latent alignment layers, **ENIGMA** achieves a ~165x reduction in parameters for multi-subject deployments, enabling efficient simultaneous inference for multiple users with less than 1% of the model size.

Motivated by the overlap between vision and mental imagery (Breedlove et al., 2020), we believe models like **ENIGMA** that target semantic content encoded in brain activity from visual perception are a meaningful step toward more general visual decoding from noninvasive brain activity, and we look forward to future research building on our work toward these goals.

## 5.1 Current Limitations

Despite training on up to 30 participants, **ENIGMA's** multi-subject training and fine-tuning approaches do not introduce any measurable performance gains, i.e., adding more subjects does not meaningfully change the performance ceiling of the model. This aligns with previous findings studying the scaling properties of neuroimaging data (Banville et al., 2025), but nonetheless represents an important limitation for the field to overcome. Our model has so far been validated only in a tightly constrained image-reconstruction paradigm, leaving its utility for more open-ended BCI tasks untested. We plan to explore the above research avenues in future work.

Like other current noninvasive EEG-to-Image methods, our reconstructions are produced from EEG averaged over repeated presentations of the same image. Single-trial decoding remains substantially poorer (Appendix A.9), and the low signal-to-noise ratio of single-trial EEG is a fundamental limitation of the medium rather than of our architecture specifically. Closing this gap, for example through stronger single-trial denoising or richer single-trial representations, is an important direction for moving toward genuinely online, single-trial brain-computer interfaces.

## 5.2 Broader Impacts

While fMRI has dominated neural image reconstruction research due to its superior spatial resolution and downstream decoding performance, its limited accessibility, high cost, and restriction to laboratory settings preclude real-world BCI deployment. **ENIGMA** demonstrates that EEG, despite its lower signal quality, provides sufficient information for semantically meaningful reconstruction when combined with appropriate architectural choices and training strategies. The convergence of rapid adaptation, hardware robustness, and computational efficiency represents a fundamental shift in the practical viability of EEG-based visual decoding, establishing a foundation for deployable brain-computer interfaces in assistive communication, clinical assessment, and real-time neural decoding applications.

## 5.3 Ethical Considerations

Research aimed at decoding cognitive states is rapidly growing in scope and capability. While these endeavors promise clear downstream benefits, they also raise serious questions about broader societal implications

and their potential for misuse. Because of these risks, we stress the importance of developing an ethical framework for the application of brain decoding devices that rigorously safeguards users' data and ensures that the technology is deployed transparently, responsibly, and for the benefit of humankind (Gordon & Seth, 2024). Our own human evaluation study was minimal risk and conducted with informed consent and fair compensation; we detail its ethical considerations, including our review status and data-handling safeguards, in Appendix A.8.2.

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

## A    Appendix

### A.1    Data Processing and Format

Raw EEG was stored in standard `.edf` files and pre-processed with MNE-Python (Gramfort et al., 2013). The Emotiv firmware applies a dual 50/60 Hz notch by default, effectively attenuating frequencies above 43 Hz, so we added a 0.5 Hz high-pass and an extra 60 Hz notch to suppress residual line noise. We epoched continuous recordings from $-200$ ms to 1000 ms relative to image onset. Synchronization glitches in the Emotiv trigger stream led us to discard 0.55–1.12% of trials, which was comparable to exclusion rates reported in earlier Emotiv evaluations (Badcock et al., 2013; Williams et al., 2020). Epochs were then baseline-corrected to the pre-stimulus window and resampled to 250 Hz to match the ATM-S benchmark (Li et al., 2024). Finally, we performed multivariate noise normalization (Guggenmos et al., 2018) where we whiten input data to improve signal-to-noise ratio (SNR). Note that we estimate the whitening matrix only on the training partition to avoid data contamination. This yielded samples $x_{\text{EEG}} \in \mathbb{R}^{N_C \times N_T}$ with $N_C = 64$ channels for THINGS-EEG2, $N_C = 32$ channels for Alljoined-1.6M, and $N_T = 250$ time points for both datasets. For multi-subject models, all models require the same channel count across all subjects, and so for these models we subsample THINGS-EEG2 to the same 32 channels present in Alljoined-1.6M. For an analysis of this step on performance, see Appendix A.7. Both the THINGS-EEG2 and Alljoined-1.6M datasets contain 4 image repetitions per training sample, and 80 repetitions per inference sample. For training and inference, we average together these multiple trials for each image presentation to further boost the SNR of the data.

### A.2    Loss function formulations

For the loss functions behind the training of ENIGMA, we use common, well-defined loss functions for mean-squared error and symmetric InfoNCE contrastive loss widely utilized in the literature, with precise mathematical definitions below.

$$L = \text{MSE}(c_{EEG}, f_{CLIP}(\text{image})) + \lambda \cdot \text{InfoNCE}(c_{EEG}, \text{norm}(f_{CLIP}(\text{image}))) \tag{4}$$

$$\text{MSE} = \frac{1}{N} \sum_{i=1}^{N} \|c_{EEG}^i - f_{CLIP}(\text{image})^i\|_2^2 \tag{5}$$

$$\text{InfoNCE} = \frac{L_{EEG \rightarrow Image} + L_{Image \rightarrow EEG}}{2} \tag{6}$$

$$L_{EEG \rightarrow Image} = -\frac{1}{N} \sum_{i=1}^{N} \log \left[ \frac{\exp(c_{EEG}^i \cdot f_{CLIP}(\text{image})^i / \tau)}{\sum_{j=1}^{N} \exp(c_{EEG}^i \cdot f_{CLIP}(\text{image})^j / \tau)} \right] \tag{7}$$

$$L_{Image \rightarrow EEG} = -\frac{1}{N} \sum_{i=1}^{N} \log \left[ \frac{\exp(c_{EEG}^i \cdot f_{CLIP}(\text{image})^i / \tau)}{\sum_{j=1}^{N} \exp(c_{EEG}^j \cdot f_{CLIP}(\text{image})^i / \tau)} \right] \tag{8}$$

Here:

- $c_{EEG} \in \mathbb{R}^{1024}$ is the output of our EEG encoding pipeline (after MLP projection)

- $f_{CLIP}(\text{image}) \in \mathbb{R}^{1024}$ is the CLIP image embedding

- $\cdot$ denotes the dot product between normalized embeddings

- $\tau = 1/\text{logit\_scale}$ is the learnable temperature parameter

- The diagonal elements $(i = j)$ represent positive EEG-image pairs

- All off-diagonal elements serve as negatives within the batch

The symmetric formulation ensures bidirectional alignment between EEG and image modalities. By maximizing $c_{EEG}^i \cdot f_{CLIP}(\text{image})^i$ for paired samples while minimizing similarity with other samples in the batch, we learn a joint embedding space capturing the semantic correspondence between brain activity and visual stimuli.

In our research we also noticed that the MSE loss term of the ATM-S backbone Li et al. (2024) was computed on unit-normalized CLIP targets: $\text{norm}(f_{CLIP}(\text{image}))$, inadvertently optimizing for angular alignment (cosine similarity) rather than Euclidean distance in the embedding space. We chose not to normalize $f_{\text{CLIP}}(\text{image})$ in the MSE component to ensure that the learned $c_{\text{EEG}}$ respects the geometry of the CLIP embedding space, and note that doing so negates the need for the secondary diffusion prior training stage in Li et al. (2024) that was previously necessary to learn the magnitude of $f_{\text{CLIP}}(\text{image})$ (Figure 4B [17]).

### A.3 Additional Details on Evaluation Metrics

We use the following image similarity metrics:

- PixCorr is the pixel-level correlation between the ground-truth images and reconstructed images.

- SSIM is the structural similarity index metric Wang et al. (2004).

- AlexNet(2) and AlexNet(5) are the 2-way comparisons (2WC) of layers 2 and 5 of AlexNet Krizhevsky et al. (2012).

- CLIP is the 2WC of the output layer of the CLIP ViT-L/14 Vision model Radford et al. (2021).

- Incep is the 2WC of the last pooling layer of InceptionV3 Szegedy et al. (2015).

- Eff and SwAV are distance metrics gathered from EfficientNet-B13 Tan & Le (2019) and SwAV-ResNet50 Caron et al. (2020) models.

For the metrics in Table 1, a two-way comparison (2WC) evaluates whether the feature embedding of the stimulus image is more similar to the feature embedding of the target reconstruction, or the feature embedding of a randomly selected "distractor" reconstruction, where the score is the percent of correctly identified target reconstructions across a pool of "distractors". Our 2WC metrics, calculated using reconstructions of the 199 other test-set stimuli as "distractors", have a notably different chance threshold from 2WC metrics presented in reconstruction papers that perform evaluations using a test set with a different number of "distractors", such as the shared1000 test set of NSD Allen et al. (2022), and are thus not directly comparable. All metrics in Table 1 were calculated and averaged across 10 images sampled from the output distribution of each method using a random seed. All metrics in Table 1 were calculated on our reproduction of other methods using their open source code, and might differ slightly from metrics reported in the original papers due to our implementation of the metrics we calculated.

#### A.3.1 Normalized average of image feature metrics

To compute the normalized average of the image feature metrics used in Fig 4 and to select the best, median, and worst reconstructions displayed in figures throughout our paper, we first standardized each metric $S_k$ to the unit interval $[0, 1]$. For metrics where lower values indicated better performance (denoted by the set $\mathcal{L}$), we applied an inverted Min-Max normalization such that the optimal raw value mapped to 1. The final score $\bar{S}_{\text{final}}$ was computed as the arithmetic mean of these normalized values, ensuring equal weighting across all image feature metrics:

$$\bar{S}_{\text{final}} = \frac{1}{K} \sum_{k=1}^{K} \left( \begin{cases} \frac{\max(S_k) - S_k}{\max(S_k) - \min(S_k)} & \text{if } k \in \mathcal{L} \\ \frac{S_k - \min(S_k)}{\max(S_k) - \min(S_k)} & \text{otherwise} \end{cases} \right) \tag{9}$$

Where:

- $K$ is the total number of metrics.

- $S_k$ denotes the raw score of the $k$-th metric.

- $\mathcal{L}$ is the set of metrics where a lower score indicates better performance.

- $\min(S_k)$ and $\max(S_k)$ are the minimum and maximum values of metric $k$ across the dataset.

## A.4 Statistical Significance of Evaluation Metrics

| Method | Low-Level | | | | High-Level | | | | Human Raters |
|---|---|---|---|---|---|---|---|---|---|
| | PixCorr ↑ | SSIM ↑ | Alex(2) ↑ | Alex(5) ↑ | Incep ↑ | CLIP ↑ | Eff ↓ | SwAV ↓ | Ident. Acc. ↑ |
| **THINGS-EEG2** | | | | | | | | | |
| ENIGMA (Multi-Subject) | ±0.0014 | ±0.0014 | ±0.15% | ±0.12% | ±0.20% | ±0.19% | ±0.0008 | ±0.0008 | ±0.89% |
| ATM-S (Multi-Subject) | ±0.0009 | ±0.0010 | ±0.20% | ±0.20% | ±0.21% | ±0.21% | ±0.0004 | ±0.0006 | ±1.15% |
| ENIGMA (Single-Subject) | ±0.0014 | ±0.0014 | ±0.14% | ±0.11% | ±0.20% | ±0.18% | ±0.0008 | ±0.0008 | ±0.87% |
| ATM-S (Single-Subject) | ±0.0013 | ±0.0013 | ±0.17% | ±0.15% | ±0.21% | ±0.20% | ±0.0007 | ±0.0008 | ±0.97% |
| Perceptogram (Single-Subject) | ±0.0014 | ±0.0015 | ±0.12% | ±0.11% | ±0.20% | ±0.20% | ±0.0007 | ±0.0007 | ±0.94% |
| **Alljoined-1.6M** | | | | | | | | | |
| ENIGMA (Multi-Subject) | ±0.0007 | ±0.0008 | ±0.14% | ±0.13% | ±0.15% | ±0.15% | ±0.0005 | ±0.0005 | ±0.77% |
| ATM-S (Multi-Subject) | ±0.0007 | ±0.0007 | ±0.14% | ±0.15% | ±0.15% | ±0.15% | ±0.0003 | ±0.0004 | ±0.82% |
| ENIGMA (Single-Subject) | ±0.0007 | ±0.0009 | ±0.14% | ±0.14% | ±0.15% | ±0.15% | ±0.0004 | ±0.0005 | ±0.78% |
| ATM-S (Single-Subject) | ±0.0007 | ±0.0008 | ±0.14% | ±0.15% | ±0.15% | ±0.15% | ±0.0004 | ±0.0005 | ±0.80% |
| Perceptogram (Single-Subject) | ±0.0008 | ±0.0010 | ±0.13% | ±0.13% | ±0.15% | ±0.15% | ±0.0004 | ±0.0005 | ±0.79% |

Table 2: Standard error measurements for evaluation metrics of EEG-to-Image reconstruction models evaluated on the THINGS-EEG2 and Alljoined-1.6M datasets. Values correspond to the standard error spread of values in Table 1 in the manuscript.

## A.5 Median and Worst-case Reconstructions

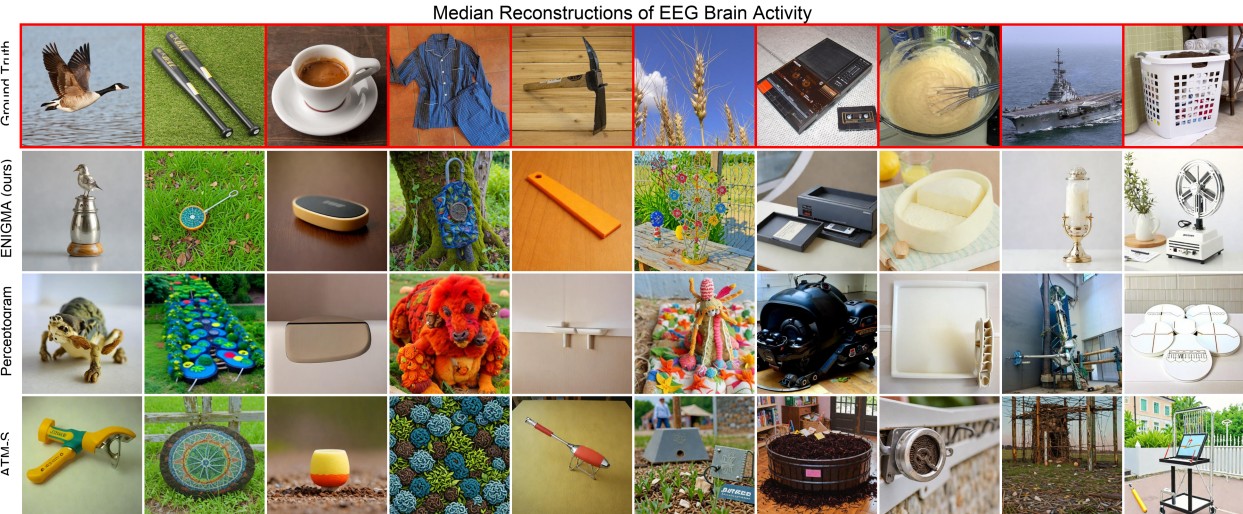

Figure 5: Qualitative comparison of reconstruction methods on seen stimuli from THINGS-EEG2. Reconstructions selected are the outputs sampled from each method and stimulus with the median scores on all of the image feature metrics in Table 1.

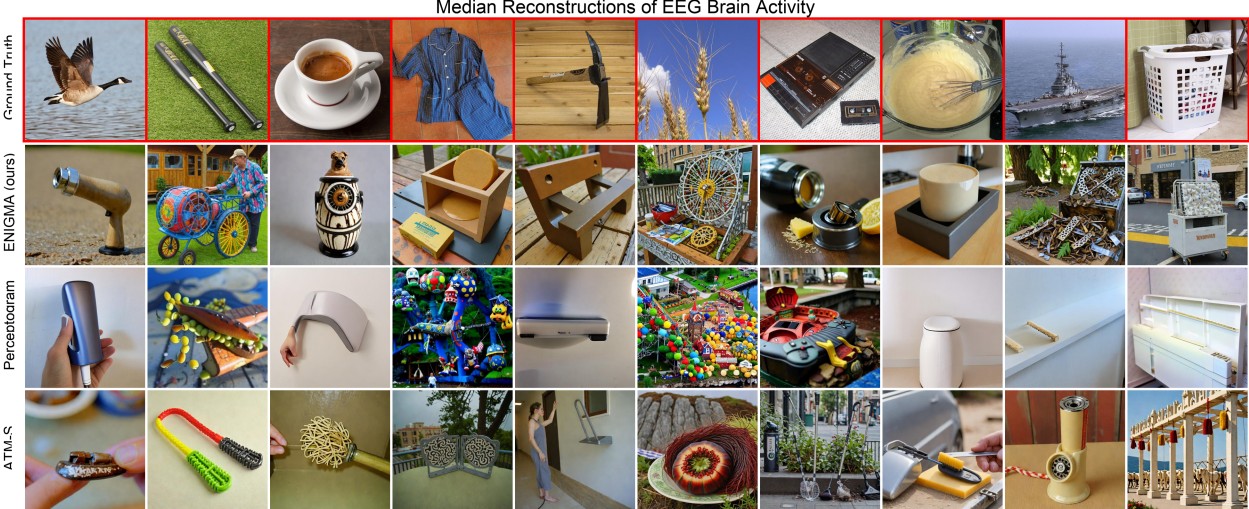

Figure 6: Qualitative comparison of reconstruction methods on seen stimuli from Alljoined-1.6M. Reconstructions selected are the outputs sampled from each method and stimulus with the median scores on all of the image feature metrics in Table 1.

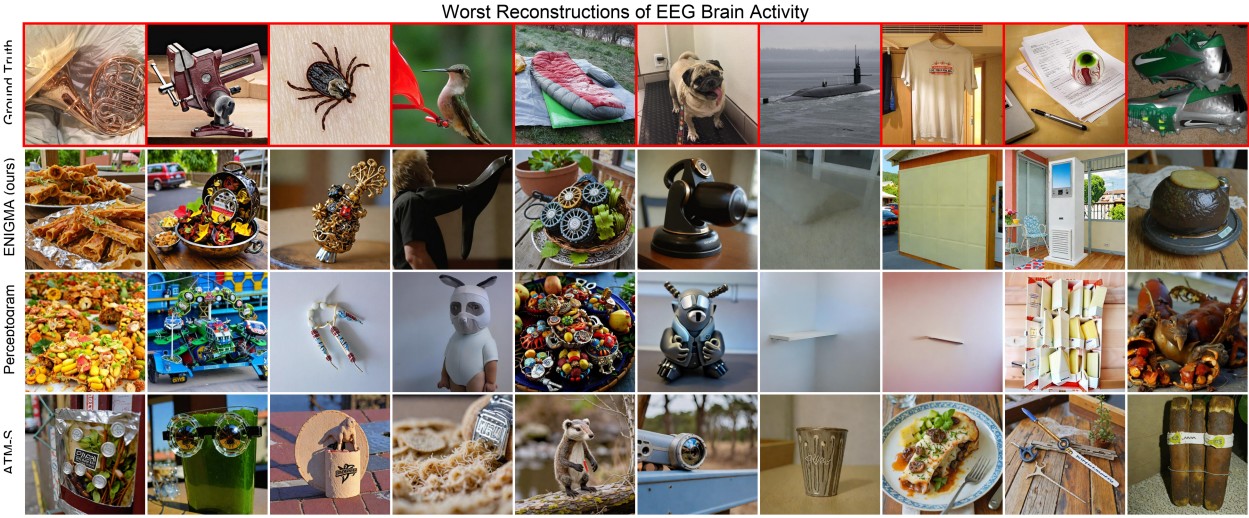

Figure 7: Qualitative comparison of reconstruction methods on seen stimuli from THINGS-EEG2. Reconstructions selected are the outputs sampled from each method and stimulus with the worst scores on all of the image feature metrics in Table 1.

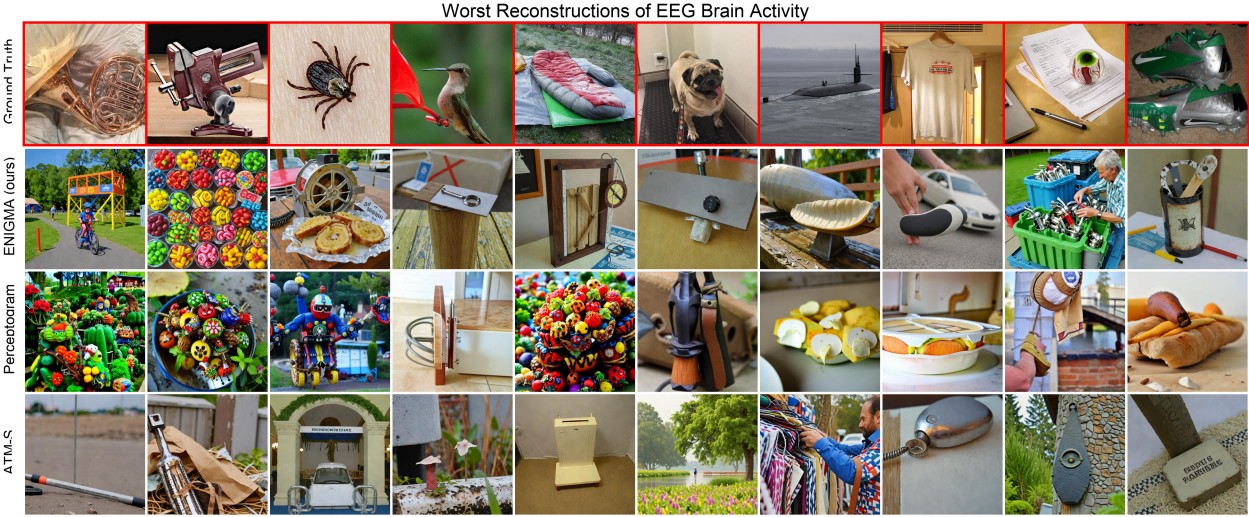

Figure 8: Qualitative comparison of reconstruction methods on seen stimuli from Alljoined-1.6M. Reconstructions selected are the outputs sampled from each method and stimulus with the worst scores on all of the image feature metrics in Table 1.

### A.6 Scaling

### A.6.1 Dataset Scaling Analysis

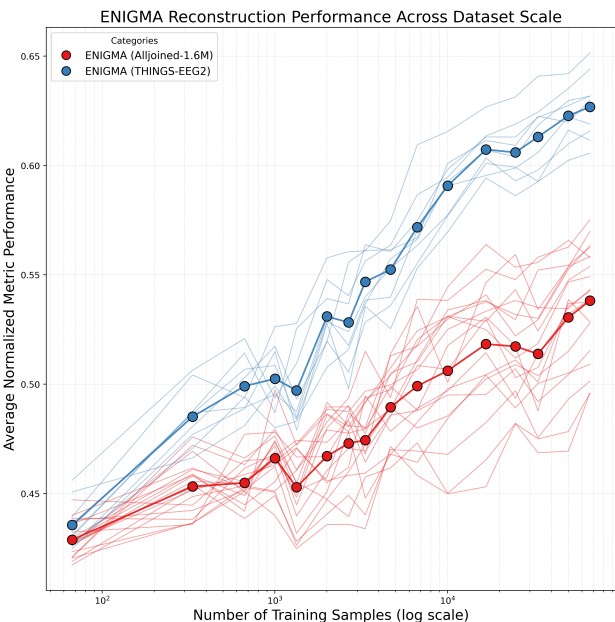

Figure 9: Scaling analysis of **ENIGMA** (not pretrained) performance on the THINGS-EEG2 and Alljoined-1.6M datasets. The number of training samples are plotted on a log-scale X-axis, and the normalized average of feature metrics presented in Table 1 is plotted on the Y-axis for each subject, with the bolded line being the average of all subjects.

When collecting large neuroimaging datasets, it is useful to be able to predict how efficiently models will scale on newly collected data. As shown in Figure 9, the reconstruction performance of **ENIGMA** increases log-linearly with the number of training samples on both available datasets, with no evident saturation on either dataset, however, performance scales with a larger exponent factor on THINGS-EEG2 that was collected on research-grade hardware. Such divergence suggests that while sheer data volume does reliably boosts accuracy, the quality of recording hardware significantly accelerates learning efficiency and leaves headroom for further gains. As highlighted with the release of Alljoined-1.6M Xu et al. (2025), this difference in scaling efficiency between EEG hardware quality is a key limitation to overcome for practical BCI applications.

### A.6.2 Pretraining Scaling Analysis

Section 4.3 and the dataset scaling analysis in Appendix A.6.1 show that multi-subject pretraining is most valuable in the low-data regime. Here we ask the complementary question: once a target subject contributes a full training set, does the number of pretraining subjects still affect final reconstruction quality? We isolate the CLIP ViT-L/14 two-way comparison (2WC) score and measure the median subject's final reconstruction quality, after fine-tuning on the complete target-subject training set (66,840 trials), as a function of the number of pretraining subjects (Figure 10).

Across the full sweep from 1 to 29 pretraining subjects the score moves only within a narrow band, from roughly 0.800 to 0.826, a total range under three points of two-way accuracy. The trend is weakly positive and non-monotonic, peaking near 16 subjects and fluctuating thereafter, and the standard errors of the averaged per-sample scores overlap across nearly all conditions. The data therefore do not support a reliable ordering of pretraining-cohort sizes at full target data: the point-to-point differences are comparable to the per-condition noise. The practical implication is that, for this architecture, the value of multi-subject pretraining is one of data efficiency rather than of raising the attainable performance ceiling once ample target data is in hand. This is consistent with the limitation noted in Section 5.1 and with prior scaling observations for neural decoding (Banville et al., 2025).

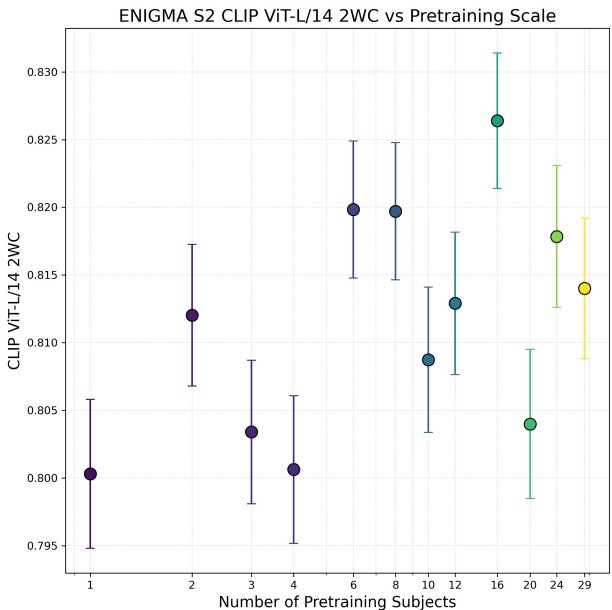

Figure 10: Final CLIP reconstruction performance for median subject S2 as a function of pretraining cohort size. Each point is the mean CLIP ViT-L/14 two-way comparison (2WC) score for subject S2's reconstructions after fine-tuning on the full target-subject training set (66,840 trials), with the model pretrained on the indicated number of other subjects (log-scaled x-axis); color encodes cohort size. Error bars denote the standard error of the per-sample scores averaged into each point (200 test images times 10 repetitions). See Appendix A.3 for a full description of the CLIP ViT-L/14 2WC metric. Differences across pretraining scales are small relative to their standard errors and largely overlap, indicating that once full target-subject data is available, the number of pretraining subjects has little effect on final reconstruction quality.

## A.7 Channel Count Ablation Analysis

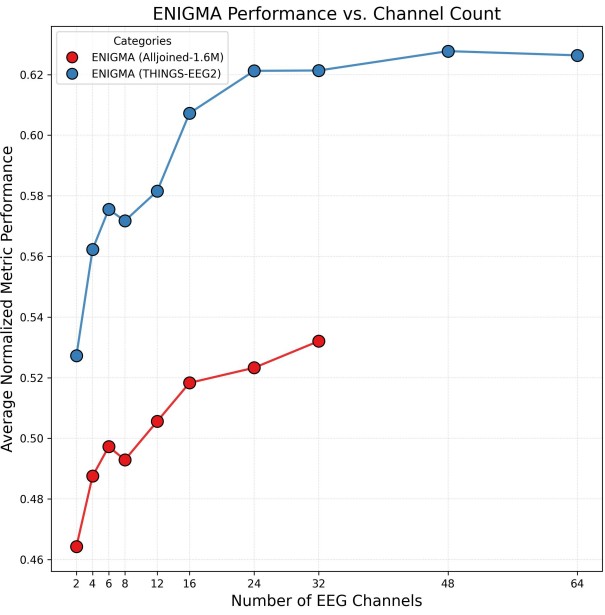

Figure 11: Channel count analysis of model performance for each dataset. The number of channels in each dataset was progressively reduced, while the remaining channels focus primarily on occipital cortex. The Y axis is plotted the same as Fig. 9.

A commonly-asked question is the number of channels needed to obtain high quality reconstructions from EEG-to-Image reconstruction models. We analyzed how the number of channels affects decoding performance using **ENIGMA**, and explored whether this contributed significantly to differences in performance between performance on the two benchmark datasets. We sub-sampled varying numbers of channels from both datasets, while retaining a focus on covering occipital cortex. We find that while performance did drop with fewer channels, channel count is not the most significant factor accounting for the performance difference between the datasets, and performance starts to drop off after 24 channels for both datasets. This suggests that it might be possible to achieve reasonable decoding performance with fewer than 32 channels.

### A.8    Behavioral Evaluation Experiments

To evaluate the quality of EEG-to-Image reconstruction models applied to our dataset, we conducted a behavioral experiment on 545 human raters online. For our experiment, we identified no risks to the human participants, and collected informed consent from all participants.

The experimental stimuli consist of image reconstructions sampled from the 30 subjects across THINGS-EEG2 and Alljoined-1.6M from all methods and cases in Table 1. The images were shuffled and 60 images were presented to each subject. We use attention checks to identify whether human raters were paying attention to the task and the instructions and dropped 8 human raters who failed at least 2 out of 8 attentions checks before analysis. An attention check presents the ground truth image as one of the candidate images and raters have to select the candidate ground truth image (as an image is most similar to itself) to pass.

Our subjects were recruited through the Prolific platform, with our experimental tasks hosted on Meadows. Each human rater was paid $1.25 for the completion of the experiment, and the median completion time was 5 minutes, resulting in an average payment rate of $15/hour. The code to reproduce our experiment can be found in our anonymized GitHub repository.

### A.8.1    2AFC Identification Task

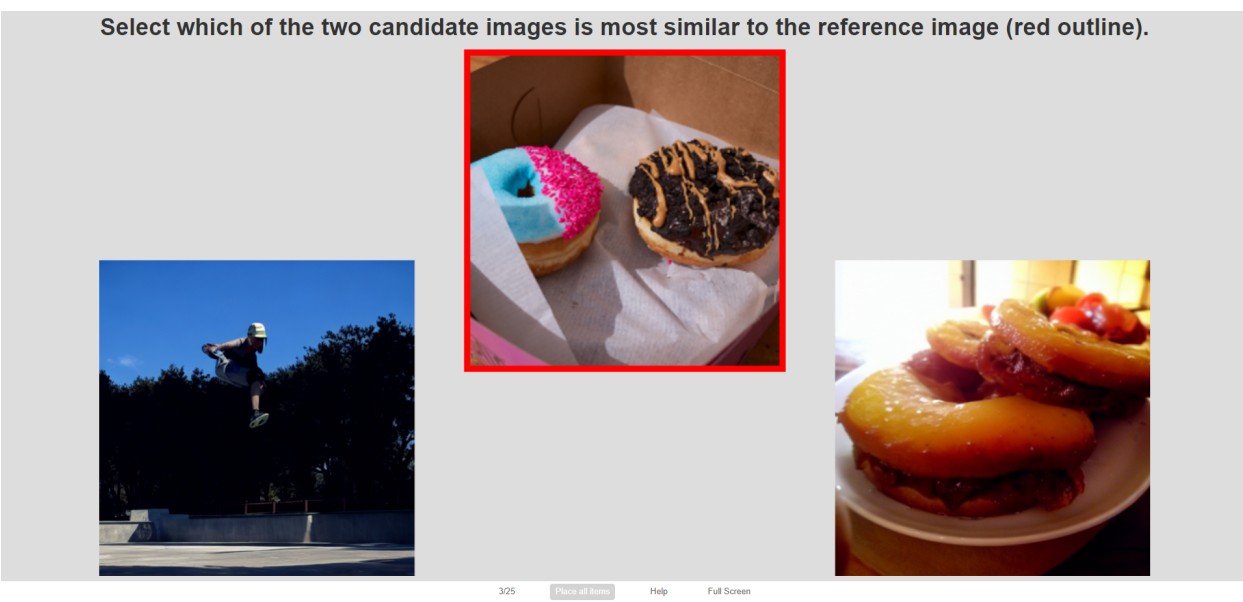

Figure 12: An example of the 2 alternative forced choice task used in our behavioral experiment performed by human raters.

Our experiment was a 2 alternative forced choice task (2AFC) facilitated by the "Match-To-Sample" task on the Meadows platform. An example of the experiment can be seen in Figure 12. In this experiment, human raters were asked to select which of two candidate images was more similar to a reference image.

The reference image provided is the ground truth image the subject either saw, and the 2 candidate images were the target reconstruction of the reference image, or a randomly selected reconstruction from an EEG recording corresponding to a different stimulus. The two candidate images were always sampled from the same reconstruction method and subject. This experiment was repeated for all reconstruction methods, model types, datasets, and subjects. With the results presented in Table 1, we establish a baseline for human-rated image identification accuracy of seen image reconstructions from EEG, as no other paper has conducted behavioral evaluations of EEG-to-Image reconstructions.

### A.8.2 Ethical Considerations

Our human evaluation (the 2AFC identification experiment described above) involved human participants. We follow the TMLR ethics guidelines and address the relevant considerations below, organized to match that page so that each point can be checked directly.

**Potential for negative societal impact.** The principal long-term concerns for neural decoding are mental privacy and potential misuse, which we discuss in Section 5.3; our method has no direct application to weapons, security or surveillance infrastructure, or the other harms enumerated in this part of the guidelines. The human evaluation reported here is itself minimal risk, and we reiterate the importance of responsible, transparent, and consent-based development and deployment of brain-decoding methods.

**General ethical conduct: human-derived data.**

1. **Personally identifiable information.** We collected no personally identifiable or sensitive personal information. Participants were identified only by the anonymized identifiers assigned by the recruitment platform (Prolific), all responses are stored in anonymized form, and individual ratings cannot be linked to a named person.

2. **Deduced information.** We did not collect demographic or protected-attribute data, and no attributes about participants were inferred, deduced, or used.

3. **Bias against groups.** The study measures whether reconstructions are perceptually identifiable, not any attribute of the raters, and no decisions about people are made from the data. Raters were drawn from Prolific's broad participant pool, and the stimuli and reconstructions are natural-object images that do not target or characterize any gender, race, sexuality, or other group.

4. **Human-subject experimentation and oversight.** The experiment is human-subjects research. It was conducted within a small private research laboratory that is not affiliated with an institutional review board (IRB) and was therefore not reviewed or approved by an external oversight board. We recognize that such review is ordinarily expected. Given the minimal-risk nature of the task (viewing images and making forced-choice similarity judgments, with no foreseeable risk beyond ordinary computer use), we proceeded under established standards for ethical human-subjects research: informed consent was obtained from every participant before the experiment began, participation was voluntary with the freedom to withdraw at any time without penalty, only adults took part, and participants were paid $1.25 for a task with a median completion time of 5 minutes (an effective rate of approximately $15/hour, above Prolific's recommended minimum). We document these safeguards here so that readers can assess them directly.

5. **Discredited datasets.** The EEG datasets used in this work, THINGS-EEG2 (Gifford et al., 2022) and Alljoined-1.6M (Xu et al., 2025), have not been discredited or withdrawn by their creators and remain publicly available.

**General ethical conduct: other data considerations.**

1. **Consent to use or share the data.** Informed consent was obtained from every rater in the user study. The EEG datasets were collected by their original authors under the consent and ethical

approvals of those studies; they comprise EEG responses to natural object images, contain no personally identifiable information, and are used here in accordance with their public release terms.

2. **Domain-specific considerations.** We recruited only adults through Prolific and did not target or knowingly include minors or other vulnerable or high-risk groups, and the task is minimal risk and not specific to any sensitive domain.

3. **Filtering of offensive content.** Stimuli are everyday object and scene photographs from the THINGS image set used by both benchmarks and contain no offensive content. Reconstructions are model-generated images conditioned on EEG, which we inspected during pilot runs and found to contain no offensive or harmful content.

4. **Compliance with GDPR and other data regulations.** Because records are keyed only by an anonymized platform identifier, a participant's data can be located and removed on request, supporting the right to erasure (for example under GDPR).

To maintain response quality we additionally included attention checks in which the ground-truth image appeared as a candidate, and excluded the 8 raters who failed at least 2 of 8 such checks prior to analysis.

## A.9 Trial Averaging

Our main protocol (Appendix A.1) averages all repetitions of each test image before decoding, a best-case condition that a single-presentation (online) deployment cannot reproduce. To characterize this dependence we measure how performance scales with the number of test repetitions averaged before inference, from the full average down to the single-trial limit. Because the question concerns the encoder rather than the generator, we score 200-way retrieval of the predicted CLIP embedding against the ground-truth image embedding (cosine similarity), which requires no diffusion stage. In Figure 13, we evaluate the full multi-subject ENIGMA model (all 30 subjects; the Table 1 multi-subject rows): for each test image we average a random subset of k of its repetitions, encode the average, and compute Top-1/5/10 retrieval, sweeping k = 1 to all 80 and averaging over 20 random subsets per k. At k = all the retrieval reproduces the published per-subject Top-1 to within rounding, anchoring the right edge of the curve to the exact Table 1 operating point and the left edge to genuine single-trial decoding.

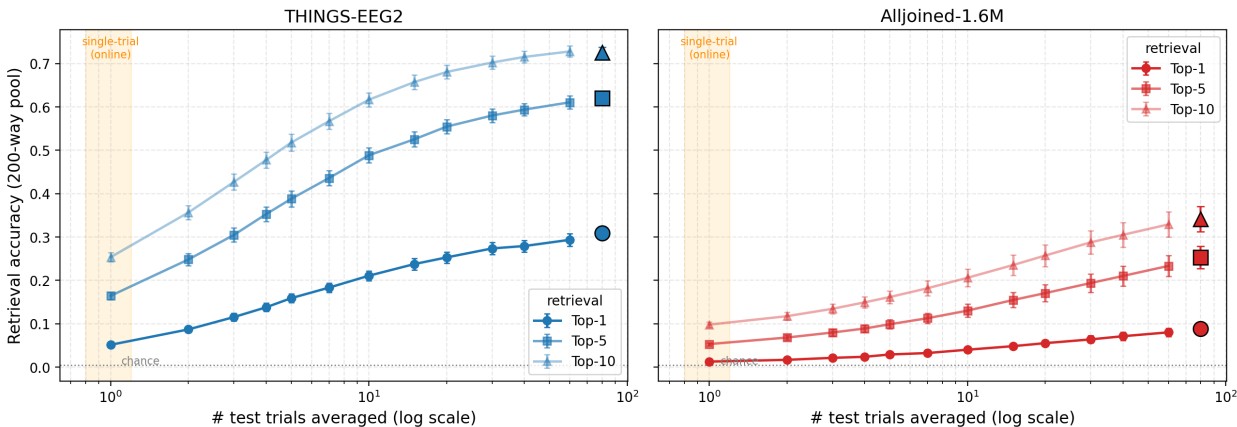

Figure 13: Retrieval accuracy (200-way pool) of the multi-subject ENIGMA encoder as a function of the number of test repetitions averaged before inference, for THINGS-EEG2 (left) and Alljoined-1.6M (right). Curves are mean ± SEM across subjects; large markers denote each subject's full 80-trial-average operating point. The shaded band at k = 1 is the single-trial / online condition; the dotted line is chance (1/200 = 0.005).

Performance degrades gracefully and monotonically as averaging is reduced, with no threshold or collapse, and single-trial decoding remains well above chance ($1/200 = 0.005$). On THINGS-EEG2, Top-1 falls from 0.309 at the full average to 0.052 at single trial (about 10x chance), Top-5 from 0.620 to 0.164, and Top-10 from 0.725 to 0.254; on Alljoined-1.6M, Top-1 falls from 0.088 to 0.013 (about 2.6x chance), Top-5 from 0.253 to 0.053, and Top-10 from 0.341 to 0.098. The encoder thus carries genuine stimulus-specific information from a single trial, but headline reconstruction quality depends on test-time repetition averaging. Single-trial online decoding is feasible but operates at the left edge of this curve.

## A.10    Embedding-level fidelity analysis

The image-similarity metrics in Table 1 are computed on the reconstructed image and therefore conflate the EEG encoder with the frozen SDXL Turbo generator, since a strong generative prior can yield a plausible image from a weak conditioning signal. To isolate the encoder we score its predicted conditioning embedding $c_{EEG}$ directly against the ground-truth embedding $f_{CLIP(image)}$, obtained by encoding the stimulus with the same frozen OpenCLIP ViT-H/14 model the decoder regresses toward (both in the same 1024-dimensional space), before any image is generated. We use the 200 held-out conditions per subject over all subjects (10 THINGS-EEG2, 20 Alljoined-1.6M) and evaluate ATM-S identically, since it shares the same target and generator. Perceptogram is excluded from this analysis because it regresses to a different CLIP target (the 257x768 token grid of ViT-L/14) and so does not occupy the same embedding space.

| Method | Cosine ↑ | Centered Cosine ↑ | RSA $\rho$ ↑ | 2-way Ident. ↑ |
|---|---|---|---|---|
| **THINGS-EEG2 (10 subjects)** | | | | |
| **ENIGMA (Multi-Subject)** | $0.452 \pm 0.002$ | $\mathbf{0.254 \pm 0.005}$ | $\mathbf{0.163 \pm 0.008}$ | $\mathbf{94.8\% \pm 0.3}$ |
| ATM-S (Multi-Subject) | $0.643 \pm 0.001$ | $\underline{0.182 \pm 0.007}$ | $\underline{0.097 \pm 0.013}$ | $\underline{63.4\% \pm 0.7}$ |
| **ENIGMA (Single-Subject)** | $0.413 \pm 0.004$ | $\mathbf{0.226 \pm 0.005}$ | $\mathbf{0.148 \pm 0.011}$ | $\mathbf{95.6\% \pm 0.3}$ |
| ATM-S (Single-Subject) | $0.341 \pm 0.005$ | $\underline{0.035 \pm 0.002}$ | $\underline{0.128 \pm 0.011}$ | $\underline{92.0\% \pm 0.9}$ |
| **Alljoined-1.6M (20 subjects)** | | | | |
| **ENIGMA (Multi-Subject)** | $0.444 \pm 0.008$ | $\mathbf{0.119 \pm 0.011}$ | $\underline{0.087 \pm 0.010}$ | $\mathbf{77.8\% \pm 2.2}$ |
| ATM-S (Multi-Subject) | $0.629 \pm 0.001$ | $\underline{0.083 \pm 0.008}$ | $\mathbf{0.096 \pm 0.008}$ | $\underline{54.8\% \pm 0.5}$ |
| **ENIGMA (Single-Subject)** | $0.373 \pm 0.010$ | $\underline{0.103 \pm 0.009}$ | $\mathbf{0.097 \pm 0.008}$ | $\mathbf{80.7\% \pm 1.9}$ |
| ATM-S (Single-Subject) | $0.625 \pm 0.004$ | $\mathbf{0.113 \pm 0.014}$ | $\underline{0.066 \pm 0.012}$ | $\underline{59.3\% \pm 1.5}$ |

Table 3: Embedding-level fidelity of the predicted conditioning embedding $c_{EEG}$ scored directly against the ground-truth CLIP embedding $f_{CLIP}$(image), before diffusion. Values are mean $\pm$ SEM across subjects (10 for THINGS-EEG2, 20 for Alljoined-1.6M); higher is better for all columns. Within each single- or multi-subject section, **bold** indicates best and underline second-best. Cosine is the mean per-sample cosine similarity between $c_{EEG}$ and $f_{CLIP(image)}$. Centered cosine is the same after subtracting the test-set mean embedding from both sides, which removes the anisotropy offset. RSA is the Spearman correlation between the predicted and ground-truth representational dissimilarity matrices (RDM = 1 - Pearson over the 200 conditions), averaged across subjects. Two-way identification is an embedding-space two-alternative forced choice (whether $c_{EEG}$ is closer to its own image embedding than to a random other, 50% chance), and is the same 2WC format of CLIP metric displayed in Table 1, but computed on the $c_{EEG}$ embedding before image reconstruction. CLIP's embedding space is anisotropic, so a constant predictor emitting the dataset-mean embedding already scores cosine 0.628; raw cosine therefore partly reflects the geometry of the space, whereas centered cosine, RSA, and two-way identification are all offset-free.

In Table 3, **ENIGMA** wins all four blocks for two-way identification, often by a wide margin (THINGS-EEG2 multi 94.8% vs 63.4%; Alljoined-1.6M multi 77.8% vs 54.8%), while the multi-subject ATM-S baseline falls toward the 50% chance line. **ENIGMA** also leads centered cosine in three of four blocks and attains the best RSA in both datasets.

In paired per-subject comparisons against ATM-S (Figure 14) (Wilcoxon signed-rank, Holm-corrected across the four settings), **ENIGMA**'s RSA is significantly higher in three of four settings and tied in the fourth

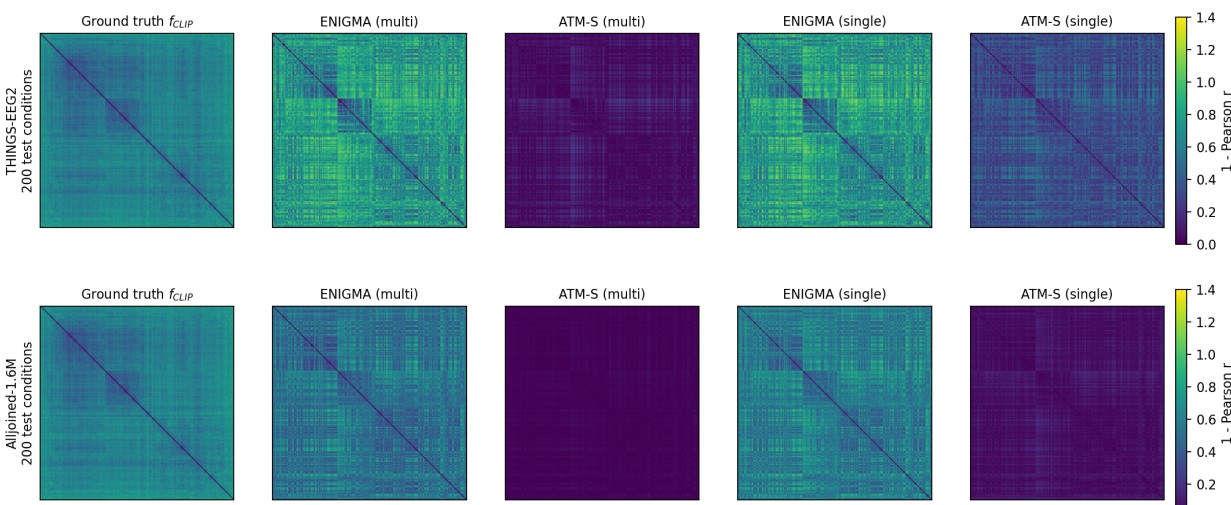

Figure 14: Representational dissimilarity matrices (RDM = 1 - Pearson) over the 200 held-out conditions, ordered by hierarchical clustering of the ground-truth RDM and averaged over subjects. **ENIGMA**'s predicted RDMs reproduce the block structure of the ground-truth CLIP RDM (first column); the multi-subject ATM-S RDM is nearly uniform, the signature of a representation collapsed toward the dataset mean.

(multi-subject THINGS-EEG2, 10/10 subjects, p = 0.008; single-subject THINGS-EEG2, 9/10, p = 0.024; single-subject Alljoined-1.6M, 13/17, p = 0.024; multi-subject Alljoined-1.6M, 6/20, p = 0.29, tied), and every model's RSA is itself above chance (condition-label permutation test, 1000 permutations, p = 0.001 for all eight model-by-dataset cells). The multi-subject ATM-S baseline collapses toward the dataset mean, which we quantify with prediction dispersion (1 minus the mean pairwise cosine among a subject's 200 predictions): **ENIGMA** 0.76 vs ATM-S 0.07 on THINGS-EEG2 (about 11x) and 0.53 vs 0.017 on Alljoined-1.6M (about 31x). This collapse is why ATM-S's raw cosine sits at the 0.628 anisotropy floor (a higher raw cosine that reflects regression to the mean, not fidelity) and why its RDM is nearly uniform (Figure 14). Measured before the generator is ever invoked, **ENIGMA**'s embedding both identifies the seen stimulus and preserves the representational geometry of CLIP space better than the baseline, so we conclude that the reconstruction quality in Table 1 is attributable to the EEG encoder and not the diffusion model.

## A.11  Fine Tuning Variance Across Subjects

Figure 4A reports the calibration time at which a pretrained ENIGMA fine-tune surpasses a fully-trained ATM-S for the median subject (S2). To test whether this 15-minute threshold is representative, we re-ran the crossover analysis per subject across all 30 subjects from THINGS-EEG2 and Alljoined-1.6M. For each subject we fine-tuned the pretrained model at 16 calibration fractions (0.1% to 100%, about 67 to 66,840 trials, 0.2 to 214 minutes) and found the smallest calibration time at which that subject's fine-tune composite reaches that same subject's fully-trained single-subject ATM-S model (the Table 1 ATM-S rows), with linear interpolation in (log trials, performance) between the bracketing fractions. The performance composite is identical to Figure 4A: the per-sample mean of the eight MinMax-normalized metrics, with EffNet-B and SwAV inverted.

We find that for all 30 subjects, ENIGMA's models surpass their corresponding fully-trained ATM-S counterparts. The overall median crossover is 19.4 minutes (Interquartile range (IQR): 4.8 to 36.4), with 16.6 minutes (IQR 8.4 to 22.1) on THINGS-EEG2 and 19.5 minutes (IQR 2.9 to 45.8) on Alljoined-1.6M. For reference, 13 of 30 subjects (43%) cross within 15 minutes, 22 of 30 (73%) within 30 minutes, 28 of 30 (93%)

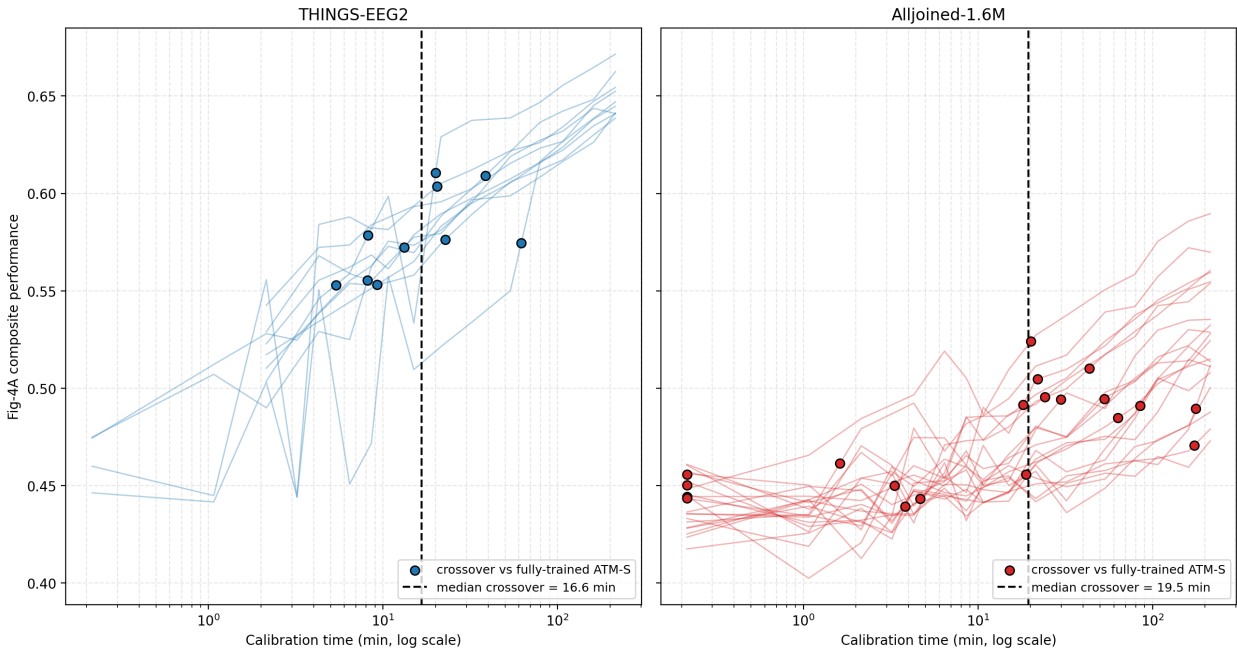

Figure 15: Per-subject calibration scaling. Each faint line is one subject's pretrained-ENIGMA fine-tune composite as a function of calibration data; each marker is where that curve crosses the subject's own fully-trained ATM-S threshold. The dashed line marks the cohort median crossover. All 30 subjects cross.

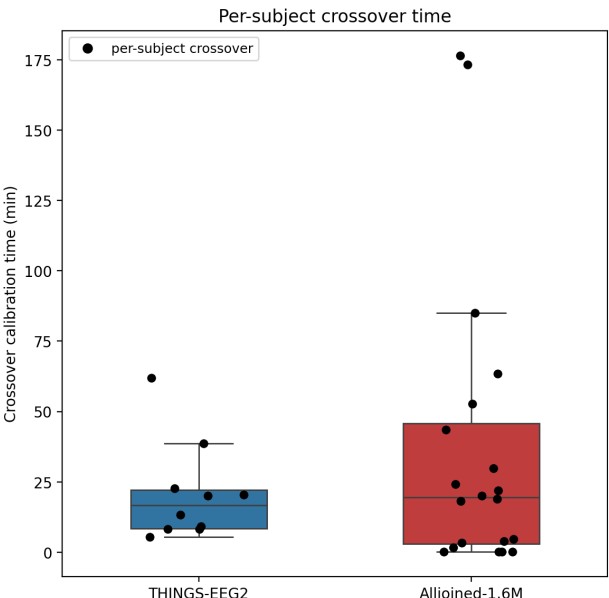

Figure 16: Distribution of per-subject crossover times (the calibration time to surpass each subject's fully-trained ATM-S), as a box plot across the 30 subjects. The 15-minute headline sits inside the interquartile range near the mean.

within 90 minutes, and all 30 by full data. The requirement is lighter on THINGS-EEG2 and heavier-tailed on Alljoined-1.6M, where two subjects (sub-20 and sub-26) cross only near full data.

