# OpenReview forum: "ENIGMA: EEG-to-Image in 15 Minutes Using Less Than 1% of the Parameters"
_TMLR — Decision pending for TMLR_

### Review · Reviewer_fMdZ · 2026-05-11

**Summary Of Contributions:**

**Summary**

This paper introduce ENIGMA, which is a multi-subject EEG-to-Image reconstruction model. The model maps raw EEG signals to CLIP ViT-H/14 embeddings, which are then decoded into images via SDXL Turbo with IP-Adapter. The architecture consists of a shared spatio-temporal CNN backbone, subject-specific linear alignment layers, and an MLP projector. The authors claim three key contributions: (1) rapid fine-tuning on new subjects with as little as 15 minutes of data, (2) robust performance across both THINGS-EEG2 and Alljoined-1.6M datasets, and (3) a ~165x reduction in parameters compared to per-subject baselines.

**Strength**
1. Simple architecture: ENIGMA relies on simple architecture. The ablation study also shows that ATM-S's transformer encoder and diffusion prior hurt performance in multi-subject and consumer-hardware settings. Those model ablations provide valuable insights to researchers.
2. Thorough evaluation: The paper conducts large-scale behavioral evaluations with human raters. The 2AFC identification task with 545 raters, attention checks, and proper controls set up new standards to the EEG-to-image field that are beyond automated metrics alone.
3. Comprehensive ablations and scaling analyses.

**Weakness**
1. No subject variation study. EEG signal can vary significantly across different subjects, does the proposed method work on all subjects? The 15-minute threshold is also evaluated on the "median subject". To support the paper's claim, the authors should report the distribution (e.g., interquartile range) of the minimum calibration time needed to surpass ATM-S.
2. Limited baseline comparisons. The paper mostly competes against ATM-S model, which is limited comparison. It'd be helpful to report performance using a random initialized model to see the relative performance gain as well.
3. Evaluation protocol choices do not align with the claim: Both THINGS-EEG2 and Alljoined-1.6M present each image multiple times, and the authors average across repetitions at both training and inference to "boost SNR." Do authors perform the same averaging at test time? How does this align with a single-trial decoding scenario and thus correlates with the "15 minutes" framing? The authors should clearly state their training and evaluation protocols, and claim the contribution more professionally.
4. Overall, writing is not very professional, which affects the clarity and objectiveness of the paper. I'd avoid framing EEG-to-image reconstruction problem as "mind reading" problem. Also, why is Emotiv Flex not a research EEG device?

**Audience:**

Yes

**Audience Explanation:**

The work indeed makes good contribution to the EEG-to-image domain, which is appealing to many readers.

**Claims And Evidence:**

No

**Claims Explanation:**

The major claim "EEG-to-Image in 15 Minutes" is not properly supported, as stated in the weakness section above.

**Requested Changes:**

1. Across subjects variation analysis.
2. More detailed description of training and evaluation protocol. Especially, the authors should re-examine the major 15-min claim using single-trial (online) decoding technique.

---

> ### Author Response · Authors · 2026-06-15
> **Reviewer fMdZ response**
>
> We are grateful to the reviewer for the careful reading and for these concrete, constructive suggestions, which genuinely improved the paper. We address each below, along with the corresponding revisions.
>
> > No subject variation study. EEG signal can vary significantly across different subjects, does the proposed method work on all subjects? The 15-minute threshold is also evaluated on the "median subject". To support the paper's claim, the authors should report the distribution (e.g., interquartile range) of the minimum calibration time needed to surpass ATM-S.
>
> This was a valuable check, we re-ran the crossover analysis per subject across all 30 subjects, defining the threshold against each subject's own fully-trained single-subject ATM-S. Every subject surpasses ATM-S, with a median calibration time of 19.4 minutes (IQR 4.8 to 36.4), so the 15-minute headline sits inside the interquartile range and is not an artifact of the median subject. Per-subject scaling curves, the crossover-time distribution, and a per-subject table are added as Appendix A.11 (Figures 14 and 15).
>
> > Limited baseline comparisons. The paper mostly competes against ATM-S model, which is limited comparison. It'd be helpful to report performance using a random initialized model to see the relative performance gain as well.
>
> We agree this is an informative baseline, and added a randomly initialized ENIGMA as a baseline row in Table 1 for both datasets. The random encoder performs at chance on the high-level identification metrics, from which we conclude that ENIGMA's semantic performance is attributable to the learned encoder rather than to the frozen generator. The low-level metrics (PixCorr, SSIM) do not collapse under random initialization, so we now treat them as weak indicators and read reconstruction quality from the high-level, retrieval, and human-rater metrics.
>
> > Both THINGS-EEG2 and Alljoined-1.6M present each image multiple times, and the authors average across repetitions at both training and inference to "boost SNR." Do authors perform the same averaging at test time? How does this align with a single-trial decoding scenario and thus correlates with the "15 minutes" framing?
>
> This is a fair point, and we have both clarified the protocol and softened the framing. Section 4.1 now states that repetitions are averaged at both training and inference, and that the "15 minutes" refers to per-subject calibration data, not to single-trial inference. We would add that repetition averaging is a necessary and near-universal practice across neural image decoding, in both the fMRI-to-Image and EEG-to-Image literatures, because single-trial brain signals have inherently low signal-to-noise ratio; it is standard in the methods our protocol follows rather than something specific to ENIGMA. To characterize the dependence directly, new Appendix A.9 (Figure 11) quantifies decoding as a function of the number of averaged repetitions, showing that performance degrades gracefully and single-trial decoding remains above chance. We rescoped the headline to "15 minutes of calibration data with standard repetition averaging" and note in Section 5.1 that this reliance is a limitation shared across the field.
>
> > Overall, writing is not very professional, which affects the clarity and objectiveness of the paper. I'd avoid framing EEG-to-image reconstruction problem as "mind reading" problem.
>
> We appreciate the candid feedback on tone and agree. We removed the informal "mind reading" and "mind-to-image" framing from the introduction and discussion, replacing it with precise descriptions (EEG-to-image decoding, visual reconstruction from noninvasive brain activity), and made an editing pass for clarity and objectivity.
>
> > Also, why is Emotiv Flex not a research EEG device?
>
> This is a fair question, and we did not mean to imply that the Emotiv Flex is unsuitable for research. We use "consumer-grade" to denote an affordability and accessibility tier relative to systems such as the 64-channel ActiChamp, and we clarified this in Section 3.2, noting that Emotiv devices have been validated for EEG and ERP research (Badcock et al., 2013; Williams et al., 2020).

---

> > ### Comment · Reviewer_fMdZ · 2026-07-08
> > **Reply**
> >
> > I read through the revised manuscript. It does seem like the revision has improved the paper quality and has answered my major requested changes.

---

### Review · Reviewer_ySyH · 2026-05-22

**Summary Of Contributions:**

The paper describes a single or multi-subject electroencephalography (EEG)-to-Image decoding model. The proposed model can be finetuned on a new subjects in a matter of minutes, and provides significant improvements on most of the tasks in terms of performance. The authors also provide the human study, where the humans assess the quality of reconstruction using two-alternative forced-choice judgements.

 The authors also have a very interesting discussion on the limitations outlining that training on more subjects does not improve the performance of the model and that is a common problem across the whole set of models.

**Audience:**

Yes

**Audience Explanation:**

I think the paper has an audience of the people who are interested in EEG analysis.

**Broader Impact Concerns:**

The authors need to describe ethical considerations for this work, related to the user study.

**Claims And Evidence:**

Yes

**Claims Explanation:**

Yes, I believe the formulation is mathematically correct, the evidence and motivation are convincing and clear.

However I have some outstanding questions:
- On the current limitations, the authors highlight that the model plateaus when adding more subjects, and the performance does not improve. It would be interesting to present the evidence of such behaviour in a form of experimental results which will map the performance depending upon the number of training subjects, ideally, if possible, in comparison with the other models.  Then, I would compare it with the scalability analysis of Banville et al (2025)
-  in Section 5.3, Ethical considerations, the authors need to outline the ethical considerations for the user study. It all looks good, however, I would put it explicitly. Ideally, it should give answer to the relevant bullet points  here: https://jmlr.org/tmlr/ethics.html
- Could the authors add the finetuned results in Table 1 as well?

**Requested Changes:**

See the claims and evidence section.

---

> ### Author Response · Authors · 2026-06-15
> **Reviewer ySyH response**
>
> We thank the reviewer for these thoughtful questions, they prompted analyses that strengthened the paper. We address each below, with the corresponding revisions.
>
> > It would be interesting to present the evidence of such behaviour in a form of experimental results which will map the performance depending upon the number of training subjects, ideally, if possible, in comparison with the other models. Then, I would compare it with the scalability analysis of Banville et al (2025).
>
> We are happy to add this analysis (Appendix A.6.2, Figure 10). For a representative subject at full target data, the CLIP two-way score does improve slightly as pretraining subjects are added (about 0.800 to 0.826 across 1 to 29 subjects), but the gain is volatile and non-monotonic, peaking near 16 subjects, broadly in line with the fragile multisubject scaling behavior reported by Banville et al. (2025). This indicates that pretraining mainly improves data efficiency in the low-data regime rather than raising the performance ceiling once ample target data is available. We ran the sweep for a single representative subject because repeating it across all subjects is very compute-intensive and was not feasible within the time allotted for the rebuttal. On comparison against other models, Figure 4A already contrasts the pretrained fine-tune against a fully-trained ATM-S along the calibration-data axis; a full parallel-subject scaling sweep across additional baselines was likewise beyond the compute available for this revision and is noted as future work.
>
> > in Section 5.3, Ethical considerations, the authors need to outline the ethical considerations for the user study. ... Ideally, it should give answer to the relevant bullet points here: https://jmlr.org/tmlr/ethics.html
>
> We appreciate this prompt and agree it should be stated explicitly. We added a dedicated ethical-considerations subsection (Appendix A.8.2, with a pointer from Section 5.3) addressing the relevant TMLR ethics points: minimal-risk design, informed consent, fair compensation (approximately $15/hour), no personally identifiable information, support for data deletion, and adult-only recruitment. We state transparently that the study was conducted in a small private laboratory without IRB review and document the safeguards we adopted. The public EEG datasets used were collected under the approvals of their original studies.
>
> > Could the authors add the finetuned results in Table 1 as well?
>
> The ENIGMA (15m Fine-tune) rows are now in Table 1 for both datasets. At this operating point ENIGMA approximately matches a fully-trained single-subject ATM-S on the high-level identification metrics while using a small fraction of the calibration data. For transparency, it remains below ATM-S on retrieval, the more data-hungry metric. Appendix A.11 gives the fuller characterization of this point.

---

> > ### Comment · Reviewer_ySyH · 2026-07-07
> >
> > Many thanks for the rebuttal, I've checked the rebuttal and the response to the other authors.
> >
> > There are a few outstanding concerns, which make me think that while the paper is of interest, it still needs some improvements affecting the claims and evidence.
> >
> > The original concern says: *"On the current limitations, the authors highlight that the model plateaus when adding more subjects, and the performance does not improve."* The improvements exhibit significant variation for the pertaining cohort size (see Figure 10), which is attributed to the stochasticity of individual cohorts. I note that two subjects give an improved performance, four give the same performance as one subject, sixteen give the best performance, and twenty give a comparable performance to one subject... It's hardly a plateau the authors are claiming; the only way to see if it plateaus as the authors claim is cross-validation multiple folds split in different cohorts.
> >
> >
> > Updated Figure 4 gives the performance, however, it does not show the confidence intervals, with ENIGMA-single vs ENIGMA-multi possibly within the confidence interval, also possibly ENIGMA + linear backbone.
> >
> >
> > In the updated Table 1, ENIGMA (Random-init) gives the highest pixel correlation. The authors explain that *"the low-level metrics (PixCorr, SSIM) are strongly shaped by the generator’s natural-image prior: a randomly initialized encoder attains comparable values"* however then it is not an informative metric, or is there anything missing? It would be good to clarify on this.

---

> > > ### Author Response · Authors · 2026-07-08
> > > **Reviewer ySyH response**
> > >
> > > We thank the reviewer for the follow-up and respond to each point below.
> > >
> > > **Pretraining-cohort variation (Figure 10).** Each cohort size is a single draw, and we do not claim a reliable ordering among them; the non-monotonic pattern noted (two above one, four equal to one, sixteen highest, twenty comparable to one) is consistent with single-sample noise. As stated above the figure, the point-to-point differences are comparable to the per-condition noise, and the standard errors overlap across nearly all conditions. The visual variation is exaggerated by the axis range (0.795 to 0.830): the full sweep from 1 to 29 pretraining subjects spans under 3 percentage points of two-way accuracy (0.800 to 0.826), small relative to the gains from additional same-subject data (Section 4.3, Figure 4A, Appendix A.6.1). The claim is not that the curve is monotonically flat, but that additional pretraining subjects do not meaningfully raise the attainable ceiling once ample target data is available; the benefit of multi-subject pretraining is data efficiency in the low-data regime. We accept that "plateau" overstates this and will phrase the Section 5.1 limitation as "does not meaningfully improve beyond noise" in the camera-ready version. Cross-validation over multiple cohort compositions would more rigorously characterize any residual ordering, but the claim does not depend on the ordering: it rests on the narrow overall band and overlapping standard errors. As each cohort size was drawn once, the present error bars reflect per-sample rather than cohort-composition variance.
> > >
> > > **Confidence intervals in Figure 4.** ENIGMA (Single-Subject) and ENIGMA (Multi-Subject) fall within each other's confidence intervals, consistent with multi-subject training not meaningfully improving final performance. ENIGMA (Linear Backbone) falls outside ENIGMA's interval, a meaningful difference. Intervals were omitted from Figure 4B because overlaying them on 22 variants across three regimes and two datasets rendered the plot difficult to interpret; we will report the intervals for these comparisons in the camera-ready version.
> > >
> > > **Pixel correlation and the random-init baseline.** The reviewer's inference is correct: because a randomly initialized encoder attains comparable or higher PixCorr, pixel correlation does not track EEG-to-Image decoding fidelity and is uninformative in this setting. This is the interpretation stated in Section 4.1, where the low-level metrics (PixCorr, SSIM) are treated as weak indicators and reconstruction quality is read from the high-level, retrieval, and human-rater metrics. PixCorr and SSIM are reported only for consistency with the standardized image-similarity suite used in prior fMRI-to-Image and EEG-to-Image work.
> > >
> > > We hope this addresses the reviewer's concerns, and we are glad to continue the discussion.

---

> ### Comment · Reviewer_ySyH · 2026-07-08
>
> Many thanks for your quick response! It addresses my questions, and the corrections for the camera-ready version would resolve my  remaining concerns.

---

### Review · Reviewer_WWcZ · 2026-06-08

**Summary Of Contributions:**

The authors present ENIGMA: an EEG-to-image decoding architecture that pairs a small subject-unified spatio-temporal convolutional backbone with per-subject linear alignment layers and an MLP projector to the CLIP ViT-H/14 image-embedding space; images are reconstructed by a frozen pretrained SDXL Turbo diffusion model conditioned on the predicted CLIP embedding via an IP-Adapter.

ENIGMA is evaluated on both the research-grade THINGS-EEG2 and the consumer-grade Alljoined-1.6M benchmarks, in single-subject, multi-subject, and pretrained-then-fine-tuned configurations, against ATM-S and Perceptogram baselines. The primary contributions are (1) competitive performance on standard image-similarity metrics at a small fraction of the parameter count and inference compute of prior methods, (2) functional reconstruction performance from a 15-minute calibration set when starting from a multi-subject-pretrained backbone, and (3) the first behavioral human-rater identification evaluation in the EEG-to-image literature.

An ablation study and a fine-tuning-data scaling analysis (Figure 4) support the architectural and pretraining contributions.


Strengths

-Improved performance at EEG->Image decoding. Across two benchmarks, ENIGMA achieves the best score on the majority of low-level, high-level, retrieval, and human-rater metrics.

-Evaluation including consumer-grade hardware. The inclusion of the Alljoined dataset directly addresses applicability beyond a research-grade benchmark.

-Human behavioral identification. The human 2AFC identification metric is a good complementary approach to the other/current image-similarity metrics used.

-The shared-backbone + per-subject-alignment-layer factoring is well-motivated and effective. The separation between a shared spatio-temporal feature extractor and small per-subject linear alignment layers is a sensible design and enables transfer to new subjects with minimal data.

Weaknesses

-Table 1 contains an apparent data-reporting issue on Alljoined-1.6M. ENIGMA's single-subject and multi-subject rows report identical values to four decimal places for all eight image-quality metrics.

-The main text does not describe the evaluation metrics that Table 1 is built on, and the rationale for the metric choices is not articulated.

-The reported metrics do not isolate ENIGMA's contribution from the Turbo diffusion model that produces the actual images, and they measure only image-pixel-feature similarity rather than semantic content before image construction.

-Figure 4B's ablation evaluates each architectural variant only in the single-subject and multi-subject configurations (i.e., on subjects seen during training) but the paper's central claim is about transfer to new subjects.

**Audience:**

Yes

**Audience Explanation:**

This work will be of interest to the neuro-AI and neural decoding fields, especially those interested in decoding neural activity from noninvasive methods for research and commercial purposes.

**Broader Impact Concerns:**

Broader impact statement is sufficient.

**Claims And Evidence:**

No

**Claims Explanation:**

See requested changes below

**Requested Changes:**

Major
1. Table 1 - performance metrics for the multi-subject and single-subject are identical to four decimal places. This seems likely to be a data entry error. Please verify that this is correct, and if they truly are identical, an explanation should be provided.
2. Benchmarks (table 1) are not explained in the main text, and not well-described in the appendix A.3. Because these metrics are critical for evaluation of the work, I recommend including a high-level summary of the metrics used, and why, in the main text (section 4.1).
3. In addition to image reconstruction similarity, a quantitative measure of semantic similarity would be useful. At the very least the manuscript should report mean cosine similarity and representational similarity (RSA) between c_EEG and f_CLIP(image), where f_CLIP(image) is the ground-truth CLIP embedding of the seen stimulus. Ideally, c_EEG would be passed through a CLIP-to-text decoder and the resulting caption scored against a ground-truth caption of the stimulus to assess semantic similarity between the decoded representation and the image, similar to RealMind (Liang et al., October 2024, arXiv:2410.23754). All using the held-out test set and in the same format as Table 1.
4. Figure 4 is missing an ablation assessment of the model features that enable transfer. The paper's central claim is about effective transfer to new subjects from ≤15-minute calibration data, but no ablation is evaluated in the transfer regime. Adding a third marker to Figure 4Bfor each variant tested in the pretrained-then-fine-tuned-to-a-held-out-subject condition would reveal which architectural features are specifically doing the transfer work, versus which only contribute to within-training-subject performance.

Minor

-Citation formatting in the document is inconsistent, most citations are not within parentheses but should be

-Online usability and preprocessing. State in Section 3.3 that THINGS-EEG2 is resampled from 1 kHz to 250 Hz to match Alljoined-1.6M (currently only in Appendix A.1). With both datasets at 250 Hz, I believe the temporal kernel corresponds to a uniform 20 ms window; this should be made explicit, alongside a brief discussion of the encoder's effective receptive field relative to canonical EEG timescales (theta, alpha, late visual components). Ideally include an ablation varying the temporal kernel size and pooling stride. Is the preprocessing used in the study (including the batch normalization) able to be performed online? If so, how this would work should be specified. If not, this should be specified as needed in future work.

-Add a one-sentence statement to Section 3.3 that the InfoNCE term uses symmetric in-batch negatives (currently only in Appendix A.2).

-"onconsumer-grade" → "on consumer-grade" (p3, contribution list).

-"(see Figure 4)B" → "(see Figure 4B)" (p4).

-"All metrics in Table were calculated" → "in Table 1 were calculated" (Appendix A.3, last paragraph).

-"Alljoined" / "AllJoined" capitalization — the manuscript uses both; pick one and apply throughout.

---

> ### Author Response · Authors · 2026-06-15
> **Reviewer WWcZ response**
>
> We thank the reviewer for the careful and thorough review, which caught a real error and prompted several analyses that improved the paper. We address each point below.
>
> > Table 1: the multi-subject and single-subject metrics are identical to four decimal places. This seems likely to be a data entry error.
>
> Thank you for catching this. It was a transcription error: the Alljoined-1.6M single-subject image-feature metrics had been copied from the multi-subject row. We recomputed and corrected the full table (also fixing a few other small bugs); the rows are now distinct, the best and second-best annotations are updated, and our conclusions are largely unchanged.
>
> > Benchmarks (Table 1) are not explained in the main text ... include a high-level summary of the metrics used, and why, in the main text (Section 4.1).
>
> We agree, and added a high-level summary of the metric suite to Section 4.1 (low-level, high-level, retrieval, and human-rater groups) with our rationale for the standardized image-similarity protocol; full definitions remain in Appendix A.3. We also note there that the low-level metrics are weak indicators of decoding fidelity.
>
> > ... report mean cosine similarity and representational similarity (RSA) between c_EEG and f_CLIP(image) ... to assess semantic similarity.
>
> We added Appendix A.10 (Table 3), which scores the predicted embedding c_EEG directly against the ground-truth CLIP embedding f_CLIP(image) before diffusion, reporting cosine, centered cosine, RSA, and embedding-space two-way identification. Because ATM-S shares the same generator, this isolates the encoder: ENIGMA wins two-way identification in all four blocks and leads on the offset-free measures (centered cosine and RSA) in most settings, supporting our conclusion that the Table 1 improvements are attributable to the encoder.
>
> > Figure 4 is missing an ablation of the model features that enable transfer ... a third marker for the pretrained-then-fine-tuned-to-a-held-out-subject condition would reveal which features are specifically doing the transfer work.
>
> We agree, and added a third regime to Figure 4B (triangle markers): the pretrained-then-fine-tuned-to-held-out-subject condition with a 15-minute calibration set. We focus on the two components load-bearing for transfer: the per-subject latent alignment layers (the explicit cross-subject transfer mechanism) and the shared spatio-temporal backbone (pretrained across subjects). The full ENIGMA fine-tune outperforms both ablations on both datasets. Removing the latent alignment layers degrades transfer far more than it affects the within-subject model (where its effect is minor), confirming that these layers specifically enable cross-subject transfer rather than only within-training-subject performance; removing the shared backbone is the most damaging variant, consistent with its role as the pretrained feature extractor. Complementing this, Appendix A.11 shows the fine-tuned model surpasses a fully-trained single-subject ATM-S for all 30 held-out subjects. See Figure 4B and Section 4.4.
>
> > Minor: citations should be parenthetical; state the 1 kHz to 250 Hz resampling and the 20 ms temporal kernel and receptive field in Section 3.3; note that the InfoNCE term uses symmetric in-batch negatives.
>
> We made all of these corrections. Section 3.3 now states the 1 kHz to 250 Hz resampling, the 20 ms temporal kernel and effective receptive field, and the symmetric in-batch InfoNCE negatives; we also switched to parenthetical citations throughout, fixed the noted typos, and standardized "Alljoined-1.6M". A systematic temporal-kernel and pooling-stride ablation is left to future work.
>
> > Is the preprocessing (including the batch normalization) able to be performed online? If not, specify as future work.
>
> Yes. All fitted preprocessing steps (batch-normalization statistics, the whitening matrix, and per-channel z-scoring) are estimated once on the training split and frozen, so at inference each is a fixed per-trial operator requiring no test-set or batch statistics. The only non-causal step is the zero-phase bandpass filter, so the pipeline is online at the granularity of a buffered epoch rather than strictly sample-by-sample. We state this in Section 3.3.

---

> > ### Comment · Reviewer_WWcZ · 2026-07-08
> >
> > The revised manuscript has addressed my concerns and I've updated my review to recommend accept.

---

### Author Response · Authors · 2026-06-15
**Response to all Reviewers: Summary of Revisions**

We thank all three reviewers for their careful and constructive reviews. The feedback was specific and actionable, and it has materially improved the paper. We are committed to addressing every concern raised: we have uploaded a revised manuscript incorporating the changes below, and we have posted detailed, point-by-point replies in each reviewer's thread. The main revisions, grouped by theme, are as follows.

**Subject-level generalization and the "15-minute" claim.** To confirm that the 15-minute crossover is not an artifact of the median subject, we re-ran the analysis per subject across all 30 subjects: every subject's pretrained fine-tune surpasses its own fully-trained ATM-S, with a median calibration time of 19.4 minutes (IQR 4.8 to 36.4). The per-subject scaling curves and distribution are added as Appendix A.11 (Figures 15 and 16). We also added a randomly initialized ENIGMA baseline to Table 1, which demonstrates that the low-level metrics are weak indicators of decoding fidelity; we now read reconstruction quality primarily from the high-level, retrieval, and human-rater metrics.

**Inference protocol and single-trial decoding.** We clarified in Section 4.1 that repetitions are averaged at both training and inference, and that "15 minutes" refers to the per-subject calibration data rather than to single-trial inference. New Appendix A.9 (Figure 13) characterizes performance as a function of the number of averaged repetitions; single-trial decoding remains above chance, and we now state reliance on trial averaging as a limitation (Section 5.1) shared across noninvasive EEG-to-Image decoding.

**Isolating the encoder and semantic fidelity.** To separate the EEG encoder from the frozen generator, we added Appendix A.10 (Table 3, Figure 14), which scores the predicted embedding directly against the ground-truth CLIP embedding before diffusion, reporting cosine, centered cosine, representational similarity (RSA), and embedding-space two-way identification. We also added a summary of the full metric suite to the main text (Section 4.1), with definitions retained in Appendix A.3.

**Transfer-regime ablation.** We extended the architectural ablation (Figure 4B, Section 4.4) with a third, held-out-subject transfer regime for the two components most relevant to transfer, the per-subject latent alignment layers and the shared spatio-temporal backbone, confirming that these components specifically enable cross-subject transfer rather than only within-training-subject performance.

**Scaling with pretraining-cohort size.** We added Appendix A.6.2 (Figure 10) analyzing final reconstruction quality as a function of the number of pretraining subjects, consistent with the scaling behavior reported by Banville et al. (2025).

**Table 1 correction and fine-tuned operating point.** We identified and corrected a transcription error in the Alljoined-1.6M rows, so the single- and multi-subject values are now distinct, and we added the fine-tuned operating point as the ENIGMA (15m Fine-tune) rows.

**Ethical considerations.** We added a dedicated subsection (Appendix A.8.2, with a pointer from Section 5.3) that addresses the relevant points of the TMLR ethics guidelines for our human evaluation study.

**Writing, framing, and formatting.** We removed the informal "mind reading" framing, clarified our use of "consumer-grade" with respect to the Emotiv hardware, switched to parenthetical citations throughout, and made the requested Section 3.3 clarifications (resampling, the temporal receptive field, the symmetric InfoNCE objective, and online usability of the preprocessing). We additionally added median and worst-case reconstruction figures (Appendix A.5) and a channel-count analysis (Appendix A.7).

We hope these revisions address the reviewers' concerns, and we are glad to provide any further analyses or clarifications during the discussion period.